# Frequency wavelength multiplexed optoacoustic tomography

Antonios Stylogiannis[1,2], Ludwig Prade[1,2], Sarah Glasl[1,2], Qutaiba Mustafa[1,2], Christian Zakian[1,2] & Vasilis Ntziachristos ●[1,2] ✉

Optoacoustics (OA) is overwhelmingly implemented in the Time Domain (TD) to achieve high signal-to-noise ratios by maximizing the excitation light energy transient. Implementations in the Frequency Domain (FD) have been proposed, but suffer from low signal-to-noise ratios and have not offered competitive advantages over time domain methods to reach high dissemination. It is therefore commonly believed that TD is the optimal way to perform optoacoustics. Here we introduce an optoacoustic concept based on pulse train illumination and frequency domain multiplexing and theoretically demonstrate the superior merits of the approach compared to the time domain. Then, using recent advances in laser diode illumination, we launch Frequency Wavelength Multiplexing Optoacoustic Tomography (FWMOT), at multiple wavelengths, and experimentally showcase how FWMOT optimizes the signal-to-noise ratios of spectral measurements over time-domain methods in phantoms and in vivo. We further find that FWMOT offers the fastest multi-spectral operation ever demonstrated in optoacoustics.

Generation of optoacoustic (OA) signals requires illumination of the sample with energy transients (e.g., pulsed or sinusoidal illumination)[1]. The sample absorbs this time-variant energy and subsequently generates an acoustic wave through thermo-elastic expansion[2]. Time Domain (TD) OA implementations offer large energy transients by means of nanosecond duration light pulses[3–6], in order to satisfy the thermal and stress confinement limits needed for optoacoustic signal generation[7]. A nanosecond duration pulse also maximizes the energy transient and optimizes the signal-to-noise ratio (SNR), thus making TD the domain of choice in optoacoustics[8–10]. TD optoacoustic imaging records the time-of-flight of the generated ultrasound waves (US) at multiple locations on the surface of the interrogated object by means of a sensitive ultrasound transducer and, using mathematical inversion, converts these measurements to three-dimensional maps of optical absorption[11].

Other imaging modalities such as optical coherence tomography (OCT) or magnetic resonance imaging (MRI) were originally demonstrated in the TD, but have benefited, in terms of imaging speed and SNR, from switching operation to the Frequency Domain (FD)[12,13]. Frequency Domain (FD) optoacoustics has been also considered as an alternative to TD, by modulating the illumination intensity at a discrete frequency and detecting the generated OA signals at the same frequency[14–16]. Signal detection is achieved with demodulation techniques that retrieve the amplitude and phase of the OA signal, a technology that is simpler and more economic than recording time signals at tens of MHz sampling rates, as is common in TD detection. FD can also enable concurrent illumination at multiple wavelengths, by modulating sources of different color at different frequencies[17–19]. Despite these advantages, intensity-modulated light[14–16] provides energy transients and corresponding optoacoustic signals that are as low as six orders of magnitude[19] weaker than the ultrashort pulses used in TD, drastically reducing the SNR in the FD[20–22]. Moreover, optoacoustic investigations at a single frequency fail to collect depth information or lead to three-dimensional imaging. We have recently shown[23] that depth information and three-dimensional image reconstruction requires the generation of signals at multiple discrete

[1]Chair of Biological Imaging at the Central Institute for Translational Cancer Research (TranslaTUM), School of Medicine, Technical University of Munich, Munich, Germany. [2]Institute of Biological and Medical Imaging, Helmholtz Zentrum München, Neuherberg, Germany. ✉e-mail: bioimaging.translatum@tum.de

frequencies, a requirement that leads to complex emission (modulation) and detection (demodulation) schemes[23,24]. Therefore, despite the potential advantages over TD[17–19,23,24], FD has had little impact in the field of optoacoustics. Frequency chirp has also been investigated as a hybrid TD-FD method, by modulating light at a continuously varying frequency[17,25], thus encoding time in frequency. Detection is carried out in the TD using time-correlation techniques. Similar to FD methods, however, the use of sine waves limits the achieved SNR, restricting the use of chirp approaches to experimental investigations.

Here, we propose Frequency Wavelength Multiplexed (FWM) operation to categorically improve upon TD performance, while minimizing FD disadvantages. FWM is effectively the reverse implementation of chirp optoacoustics, using a train of discrete pulses, similar to the ones employed in TD optoacoustics, but processing the resulting discrete frequencies that appear in the FD[23], a response that is analogous to the frequency comb appearance seen in spectral frequencies. As shown herein, FWM illumination also allows concurrent illumination at multiple wavelengths without increasing the imaging time and yielding an SNR gain that increases by the square root of the number of wavelengths N employed, over TD systems. Following theoretical considerations, we hypothesized that overdriven laser diodes[26], which offer a cheaper and more practical alternative to solid-state lasers, could exploit FWM pulse trains and lead to high-quality optoacoustic imaging that could demonstrate benefits over TD implementations. We introduce a frequency wavelength multiplexed optoacoustic tomography (FWMOT) system that utilizes 4 concurrently pulsing overdriven laser diodes, each utilizing a slightly different repetition rate in order to encode different wavelengths. These wavelengths appear then at different discrete frequencies in the frequency domain. We show concurrent multi-wavelength mesoscopic[27] imaging of lymphatic and microvascular dynamics in mice at high SNRs, offering the fastest multi-wavelength illumination ever achieved in the field of optoacoustics and confirming spectral performance that improves upon TD implementations.

## Results

### Frequency Comb Optoacoustics using a single excitation wavelength

In conventional TD operation, a square light pulse of duration $t_p$ (Fig. 1a) yields a continuous frequency spectrum in the FD, via the Fourier Transform, with the first node at the $1/t_p$ frequency. In FD operation, light modulated by a sine wave in the TD (Fig. 1b) yields a single discrete frequency in the FD. Frequency Wavelength Multiplexed modulation instead considers a train of pulses (Fig. 1c) with a pulse width of $t_p$ and a repetition rate $f_{rep}$. The Fourier Transform of this pulse train yields many discrete frequencies with an envelope (Fig. 1c, right) identical to the continuous spectrum of a single pulse with duration $t_p$ (Fig. 1a, right). The discrete frequencies of the pulse train are harmonics of the fundamental repetition rate, $f_{rep}$, i.e. integer multiples of $f_{rep}$. A higher repetition rate increases the number of pulses in the TD and reduces the discrete peaks in FD (Fig. 1d). In the FD, a longer pulse width causes the first node (at $1/t_p$) to appear at lower frequencies (Fig. 1e), reduces the overall energy density at higher frequencies, and lowers the spatial resolution[28].

Experimental validation of the FWM scheme was performed by exciting a black varnish layer on a petri dish using only L1 (Fig. 1f, g). Averaging in TD (Fig. 1g, h) increases the SNR by a factor of $\sqrt{N_p}$, where $N_p$ is the number of pulses in the pulse train. In contrast, the proposed FWM method selects the fundamental frequency $f_{rep}$ and its harmonics, $k * f_{rep}$, with $k$ being an integer, (Fig. 1i) by performing the following operation (see Supplementary Note 1):

$$S_a(\omega) = S_r(\omega) \sum_{k=1}^{N_h} \delta(\omega - k\,\omega_0), \qquad (1)$$

where $S_a(\omega)$ is the Fourier Transform of the averaged signal, $S_r(\omega)$ the Fourier Transform of the recorded signal (Fig. 1f), $N_h$ the number of the harmonics in the detected ultrasound transducer bandwidth and $\omega_0 = 2\pi/T$. The OA signal is generated at the exact same frequencies that the excitation LD pulse train contains. Therefore, operation (1) selects only the harmonics of $f_{rep}$ and filters out frequencies that do not contain signal to increase the SNR by the same factor $\sqrt{N_p}$ as in the TD (Fig. 1j). The signal in Fig. 1j is the Fourier Transform of the signal in Fig. 1h, with the two signals matching perfectly (Fig. 1k). This analysis confirms that FWM illumination results in a practical generation of multiple discrete frequencies, required for accurate FD operation, offering an SNR that is equivalent to the TD when a single wavelength is used. FWM analysis is, therefore, the Fourier space equivalent to normal time averaging. Next, we show however that FWM offers advantages over TD when multiple wavelengths are employed.

### Frequency Comb Optoacoustics using multiple excitation wavelengths

To demonstrate the FWM advantage over the TD (Fig. 2) we plotted the single-wavelength excitation pattern in the TD and its power spectrum in Fourier space (Fig. 2a) to serve as reference for the analysis that follows. The pulse train shown has a period $T$ that corresponds to a repetition rate $f_{rep} = 1/T$, a total number of pulses $N_p$, a pulse width $t_p$ and an acquisition time $t_{acq} = N_p \times T$. The period $T$ defines the maximum depth-of-view $DoV = v_s \times T$ that can be achieved for the pulse train selected, where $v_s$ is the speed of sound.

TD wavelength multiplexing is performed using wavelength interleaving, or time-sharing. However, when increasing the number of wavelengths in the TD, at least one of the following three parameters has to be compromised: the $DoV$, the number of pulses in the pulse train and therefore the SNR for each wavelength, or the total acquisition time. Figure 2b shows how the DoV is reduced when using four wavelengths at a given total acquisition time. The different wavelengths excite the tissue using the same repetition rate $f_{rep}$ but with a time shift $t_{sh}$ between each wavelength (Fig. 2a), given by $t_{sh} = T/N$, where $N$ is the number of wavelengths. The result is a reduction of the time between subsequent pulses, limiting the $DoV$ available to each wavelength by a factor $N$. Alternatively, it is possible to retain the original $DoV$ by dropping the repetition rate for each wavelength, $j$, to $f_{rep,j} = f_{rep}/N$ and the number of pulses per wavelength to $N_p/N$ (Fig. 2c); however, this approach results in an SNR reduction by a factor of $\sqrt{N}$. A third alternative retains the original $DoV$ and SNR while prolonging the acquisition time by a factor $N$ (Fig. 2d).

In contrast to TD, FWMOT employs a small frequency shift $\delta f$ for the repetition rate of each wavelength (Fig. 2e; time), whereby $\delta f \ll f_{rep}$. Each wavelength has a different repetition rate, which results in a slightly different effective $DoV$; however, since $\delta f \ll f_{rep}$, this difference is insignificant. The power spectrum of the FWMOT excitation pattern (Fig. 2e; right) shows the appearance of harmonics from the fundamental repetition rate for each wavelength. The repetition rate of the first laser, $f_{rep,1}$, can be selected as the reference repetition rate, with the repetition rate of the remaining lasers given by $f_{rep,j} = f_{rep,1} + (j-1) * \delta f$. FWMOT thus recovers the OA signals by evaluating Eq. (1) with the corresponding $\omega_0 = 2\pi f_{rep,j}$ for each wavelength. Consequently, using frequency separation, FWMOT can multiplex different wavelengths without compromising the $DoV$, SNR or acquisition time. With the signal processed in the FD, the frequency resolution is defined by $df = 1/t_{acq}$, where $t_{acq} = N_p/f_{rep,1}$, meaning that frequencies that differ by less than $df$ cannot be resolved. In order to recover the OA signal from all wavelengths, all harmonics from all lasers that lie in the Ultrasound Transducer (UST) detection bandwidth should therefore be spaced at a distance exceeding the frequency resolution $df$, imposing a minimum number of pulses $N_{p,min}$ that depends on the UST bandwidth limits ($f_{low}$ and $f_{high}$), $N$ and $f_{rep,1}$ (see Supplementary Note 2). The parameters $N$, $f_{low}, f_{high}, f_{rep,1}$ and the chosen number of pulses, $N_p$, define the range of

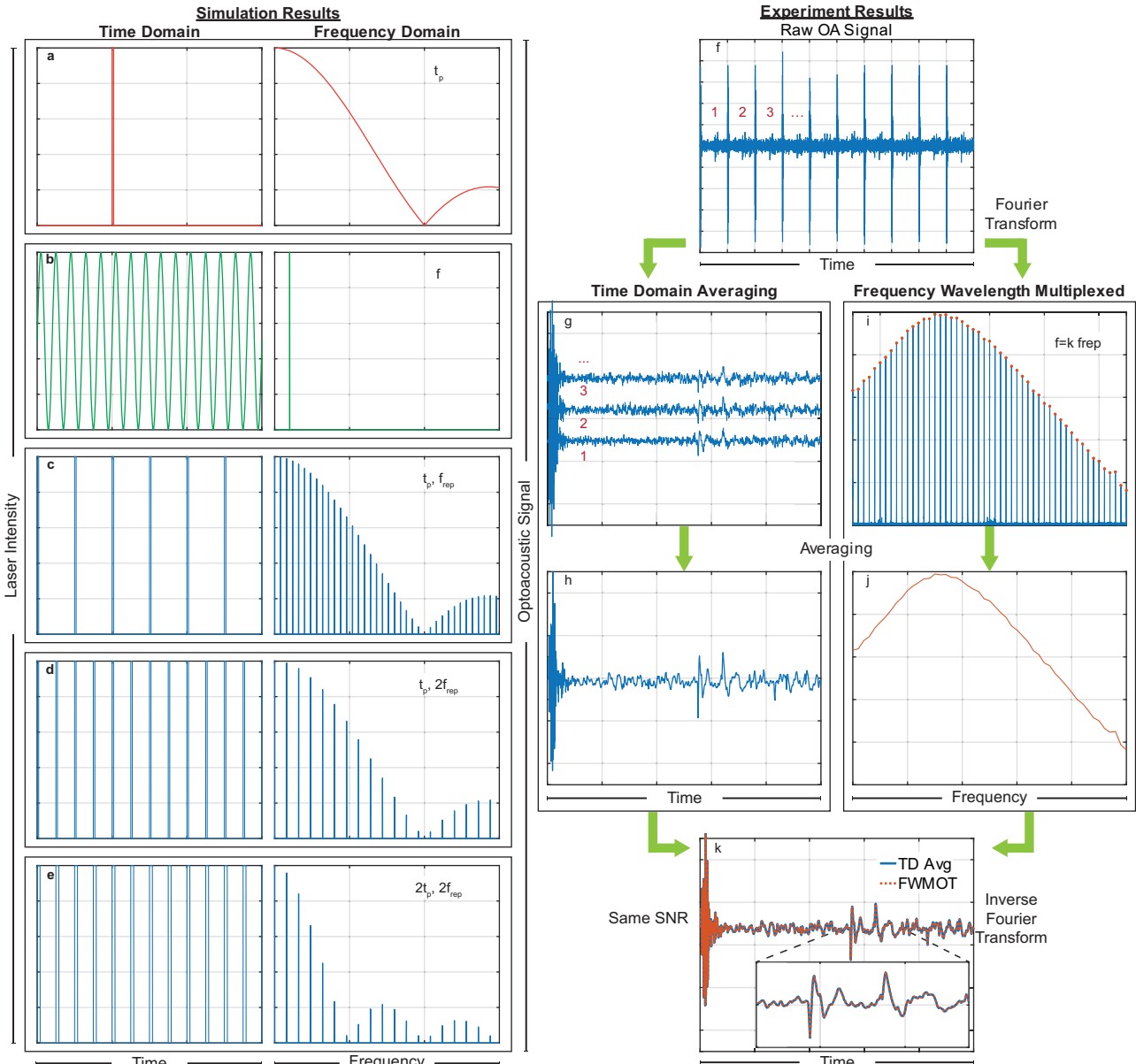

**Fig. 1 | The signal processing algorithms used in Frequency Wavelength Multiplexed Optoacoustics at a single wavelength. a** A single excitation light pulse of $t_p$ duration in TD (left) and a continuous spectrum of frequencies in FD (right). **b** A sine wave of frequency $f$ in TD and FD, continuous wave in TD and a single discrete peak in FD. **c** A train of pulses with pulse duration $t_p$ and repetition rate $f_{rep}$ in TD and FD. Many discrete pulses in TD and many discrete frequencies in FD with the same envelope as a single pulse of $t_p$ duration in **a**. **d** A train of pulses with pulse duration $t_p$ and repetition rate $2f_{rep}$ in TD and FD. Twice as many pulses in TD but half the number of discrete frequencies in FD compared to **c**. **e** A train of pulses with pulse duration $2t_p$ and repetition rate $2f_{rep}$ in TD and FD. As many pulses in TD and discrete frequencies in FD as in **d** but now following a different envelope than (**a**) or

(**c**). **f** The raw optoacoustic signal recorded using a pulse train like (**c**) for example with 1, 2, 3 and … indicating the different periods. **g, h** present the normal averaging in TD. **g** The train of pulses is split in sections of period $T = 1/f_{rep}$, indicated by 1, 2, 3 and … which are averaged (**h**) point by point. **i, j** The Frequency Wavelength Multiplexed processing of the same signal. **i** The Fourier transform of the raw optoacoustic signal (**f**) with many discrete frequencies that are all harmonics ($k * f_{rep}$ with $k$ positive integer) of the base repetition rate $f_{rep}$. In FD we choose only the harmonics of $f_{rep}$ and discard all the other frequencies that contain only noise (**j**). **k** By performing the inverse Fourier Transform in **j** we recover the TD signal that matches perfectly with the one in **h**.

small frequency shifts ($\delta f$), between $\delta f_{min}$ and $\delta f_{max}$, required to recover the signal of each wavelength without losses. A $\delta f$ value larger than $\delta f_{min}$ ensures that harmonics of laser $j$ and $j + 1$, which lie on the lower end of the UST bandwidth, are well resolved, while a $\delta f$ value smaller than $\delta f_{max}$ ensures that harmonics of laser 1 and $N$, which lie on the upper end of the UST bandwidth, are well resolved.

## FWMOT outperforms TDOA in SNR, DoV or imaging speed

To experimentally demonstrate the advantages of FWM operation in multi-wavelength excitation compared to TD optoacoustics, we

employed FWMOT with all four wavelengths. The resulting optoacoustic signal at wavelength 1, $f_{rep,1} = 200$ kHz and $N_p = 200$ is presented in Fig. 2f and attained $t_{acq} = 1$ ms and DoV = 7.5 mm. Electromagnetic interference from the laser diode (LD) driving circuit is indicated with $\alpha$. The primary OA signal generated from black varnish on a petri dish is indicated with $\beta$, whereby reflections from the UST glass lens that arrive later in time are indicated with $\gamma$. The "Noise Level" inset shows a separate measurement of the noise level that yielded a standard deviation of 90.6 µV. The intensity of the OA signal from LD1 was measured to be 12 mV, resulting in an SNR of 21.2 dB.

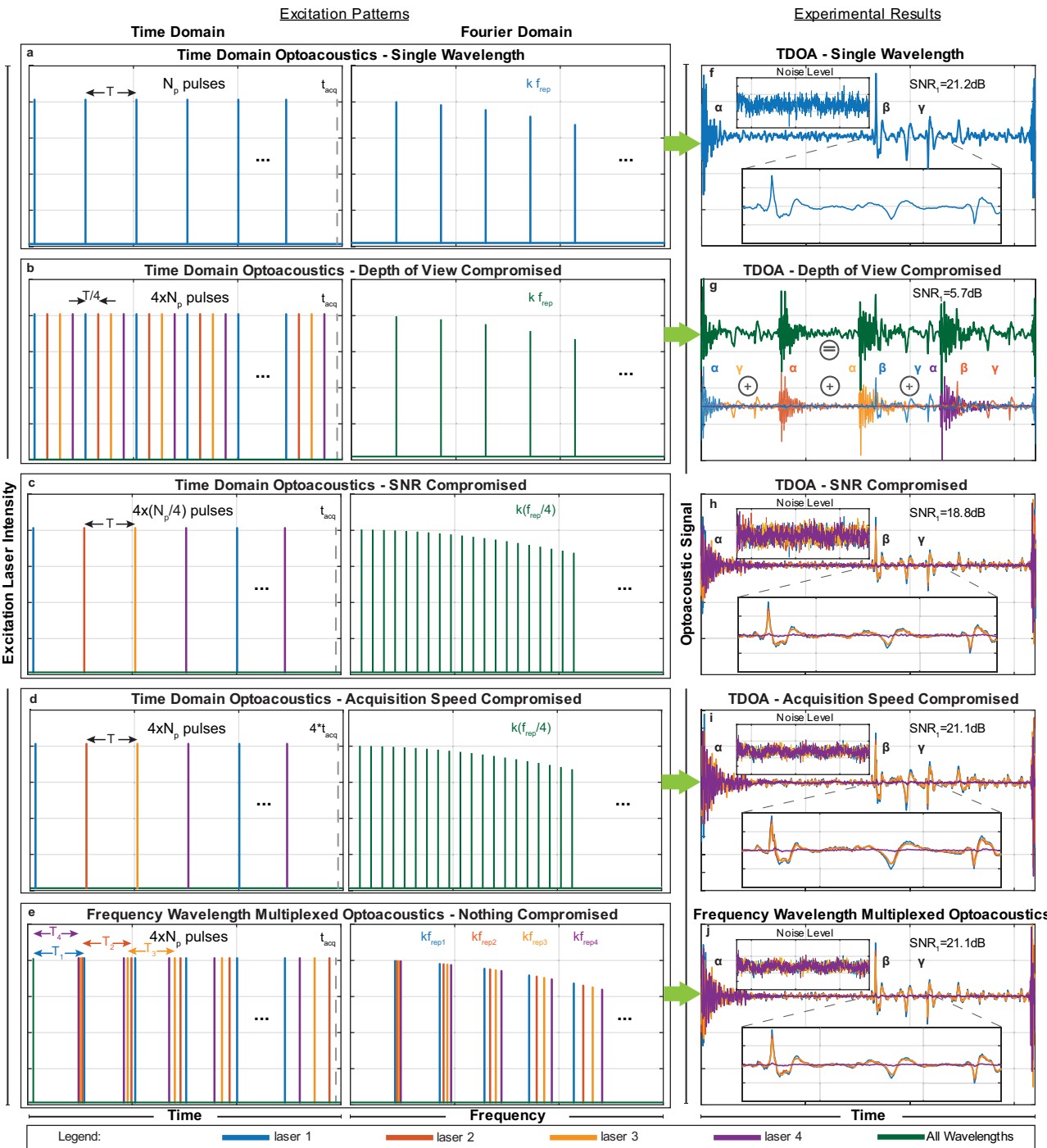

**Fig. 2 | Frequency wavelength multiplexed optoacoustic tomography (FWMOT) advantages at multiple wavelengths. a** A pulse train of one wavelength with period $T$ and repetition rate $f_{rep} = 1/T$, with $N_p$ pulses and acquisition time $t_{acq}$ in time domain (TD, left) and Fourier domain (right), with $k\,f_{rep}$ the harmonics of $f_{rep}$ with $k$ an integer. **b–d** Multiple wavelength excitation in TD Optoacoustic (OA). **b** The excitation pattern of 4 wavelengths emitting at the same repetition rate $f_{rep}$ with a time shift $T/4$, $N_p$ pulses for all wavelengths and $t_{acq}$ acquisition time. **c** The excitation pattern of 4 wavelengths with $f_{rep}/4$ repetition rate, $N_p/4$ pulses for each wavelength and $t_{acq}$ acquisition time. **d** The excitation pattern of four wavelengths with $f_{rep}/4$ repetition rate, $N_p$ pulses per wavelength but with $4t_{acq}$ acquisition time. **e** FWMOT excitation where all four wavelengths have different repetition rates $f_{rep,1}$, $f_{rep,2}$, $f_{rep,3}$, $f_{rep,4}$, $N_p$ pulses for each wavelength and $t_{acq}$ acquisition time. **f–j** The OA signal recorded by a black varnish layer on a petri dish from the excitation patterns in (**a–e**) respectively. The signal-to-noise ratio (SNR) and noise level are inset for all cases. **f** α, the electromagnetic interference from the laser diode circuitry when triggered, β the OA signal from the black varnish, γ the reflection of the OA signal in the petri dish or in the acoustic lens of the Ultrasound Transducer. **g** The OA signal from the excitation pattern **b** for all wavelengths (top line) is the sum of the OA signal from each wavelength (bottom line), with drastically reduced Depth-of-View (DoV) for each wavelength. The laser interference, OA signal and its reflections (α, β, γ) for each wavelength are indicated. **h** The OA signal at each wavelength has reduced SNR. **i** The OA signal has the same SNR but with increased acquisition time. **j** Signal from all four wavelengths has been recovered without any cross-talk between the lasers and correctly co-registered in time without compromising DoV, SNR or acquisition time. Blue, red, orange, purple, and green are used to indicate laser 1, laser 2, laser 3, and laser 4, respectively.

Figure 2g presents the averaged OA signal in the TD, when the DoV is compromised. Each LD has $f_{rep,j}$ = 200 KHz, $N_p$ = 200 and a time shift between each other that results in DoV = 1.875 mm and $t_{acq}$ = 1 ms. The OA signal resulting from the simultaneous excitation using the pattern in Fig. 2b is presented as a green line and is the sum of the individual OA signals obtained when each wavelength was pulsed separately (Fig. 2g). We could easily detect the laser trigger interference for all wavelengths (α). The OA signals from wavelength 1 and 2 (β) are located very closely to the laser interference (α) of wavelengths 3 and 4 respectively, vastly reducing their SNR (5.7 dB for wavelength 1). However, the OA signal of wavelength 3 is completely masked by the interference of wavelength 1. The reflections of the OA signal from wavelength 1, 2 and 3 (γ) are still visible. Therefore, electromagnetic interference and OA reflections further limit the DoV and SNR achieved in multiple wavelength TD optoacoustics.

Likewise, SNR limits are imposed (Fig. 2h) in response to an excitation pattern (Fig. 2c) that uses $f_{rep,j}$ = 50 kHz, $N_p$ = 50 and DoV = 7.5 mm for each wavelength with $t_{acq}$ = 1 ms. For all wavelengths, we observed that the laser interference (α), the OA signal (β), and the reflections (γ) are all visible but with lower SNR. In this case, the noise standard deviation (presented in the Figure insets) was 161.6 μV, which is higher than the 90.6 μV measured in the pulse train with 200 pulses to average (Fig. 2f), while the signal intensity remained 12 mV resulting in an SNR of 18.8 dB for wavelength 1. Finally, with $f_{rep,j}$ = 50 kHz, $N_p$ = 200, and DoV = 7.5 mm for each laser, the OA signal of all 4 lasers can be recorded without SNR losses ($N$ = 90.6 μV and $S$ = 12 mV with 21.2 dB for wavelength 1) but with a longer acquisition time, $t_{acq}$ = 4ms, that is presented in Fig. 2i.

Conversely, FWMOT operation for four wavelengths uses $f_{rep,1}$ = 200 kHz, $\delta f$ = 125 Hz and $N_p$ = 200 and recovers each signal without cross-talk between wavelengths (Fig. 2j), experimentally confirming theoretical predictions. Supplementary Figs. 4 and 5 present the Laser Diode excitation time series, the corresponding OA signal time series and the power spectra of both, in order to further showcase FWMOT's operation and verify the method's ability to recover each signal without cross-talk (see Supplementary Note 3). FWMOT is able to provide high SNR for concurrent excitation with multiple wavelengths, without extending the acquisition time (1 ms) and achieving the same DoV (7.5 mm) and SNR (here again $N$ = 90.6 μV and $S$ = 12 mV, resulting in an SNR of 21.2 dB for wavelength 1).

We would expect an SNR increase of 3 dB upon increasing the numbers of averaged measurements from 50 to 200. However, we measured an SNR increase of 2.3 dB for both TD averaging and FWM. This discrepancy can be attributed to the presence of systematic electromagnetic noise in the system, which is also averaged and reduces the SNR by 0.7 dB. However, this effect is a general feature of the OA system and not a disadvantage of the FWM algorithm.

We also compared the SNR obtained in conventional FD optoacoustics to FWMOT by employing the same black varnish phantom and single-wavelength illumination at 445 nm (see Supplementary Fig. 1). FD optoacoustics employed a sine wave of 20 MHz frequency with adjusted mean power to equal the mean power output of the FWMOT pulsed pattern used for 6.8 ns pulses at 200 KHz. FWMOT demonstrated an SNR that was 20.8 dB higher compared to FD excitation.

## FWMOT multi-wavelength imaging of tissues and tissue dynamics in vivo

While the theoretical merits of FWM optoacoustic operation were demonstrated with phantom measurements, a next critical step was to examine whether FWM could offer realistic implementations. For this reason, we aimed to investigate whether the theoretical advantages could lead to operational characteristics (SNR, acquisition speed) that would render FWMOT appropriate for in vivo applications. A particular unknown parameter in this interrogation was the FWMOT performance achieved with multiple wavelengths using laser diodes, as it

would be impractical to implement FWMOT with multiple solid-state lasers due to cost and size. To examine the merits of using low-cost technology, we investigated the performance of multiple laser diodes to image vasculature and lymphatics in vivo, using the mouse ear as a model. This imaging target was selected because it is a typical tissue on which TD optoacoustic implementations have been conventionally demonstrated.

First, we assessed whether FWMOT could produce high-quality images from biological specimens. We employed FWM illumination at 445 nm and 465 nm and resolved oxy- and deoxy- hemoglobin (Fig. 3a, b) based on their spectral difference, with de-oxygenated hemoglobin absorbing higher at 445 nm, and vice-versa. The data were collected on a grid and were first averaged point-by-point using the FWM algorithm before being fed into the reconstruction algorithm (see methods). Superposition of Fig. 3a, b revealed a color-coded composite image (Fig. 3c) of the relative vascular oxygenation, with red color corresponding to higher oxygenation levels. We further confirmed that FWMOT can produce depth-resolved images (see Supplementary Fig. 2) without cross-talk between the wavelengths, offering first evidence that FWMOT can acquire images from biological tissues based on LDs.

We next aimed to identify whether FWMOT could visualize multiple moieties in tissue without compromising operational characteristics as in the TD. We introduced exogenous contrast by intradermal injection of Evan's Blue and Indocyanine Green (ICG) and applied 4-wavelength FWMOT to simultaneously resolve arteries and veins (Fig. 3d, e) and lymphatic vessels revealed by contrast enhancement (Fig. 3f, g). Figure 3h shows the corresponding bright-field image of the mouse ear. The composite image of four wavelengths (Fig. 3i) enabled visualization and co-localization of the vascular and lymphatic vessels. Notably, the use of an ultra-wideband transducer enabled to resolve the fine structures represented by the vessels as well as the large absorbing areas that were formed around the injection sites. FWMOT can therefore be effectively used to perform OA imaging with multiple wavelengths simultaneously in vivo with an acquisition of ~30 min, while the same multispectral implementation in TD requires ~2 h, rendering such a measurement impractical.

The FWM acquisition acceleration over TD demonstrated in Fig. 3 also points to an FWMOT use for detecting rapid changes simultaneously using multiple wavelengths. We therefore applied FWMOT to monitor oxygenation fluctuations in the mouse ear during an oxygen stress test in vivo. Figure 4a shows the composite OA image from 2 blue wavelengths revealing the oxygenated and de-oxygenated vessels. We selected a 2 mm line (Fig. 4a; blue box) to acquire signals repeatedly (continuous FWM operation) at a rate of ~4 Hz. A B-scan (indicated by the letter (i) on Fig. 4a) revealed an artery (red) and a vein (green) as a function of depth. The oxygen challenge was provided by alternating the composition of the breathing gas from 0.8 liters per minute (lpm) of 100% $O_2$ (Fig. 4b; "$O_2$") to 0.6 lpm of 20% $O_2$ plus 0.2 lpm $CO_2$ (Fig. 4b; "Air"). Figure 4b plots the ratio of the wavelength 2 signal ($S_2$) over the wavelength 1 signal ($S_1$) over time. The $S_2/S_1$ ratio is indicative of the relative changes of oxygenation in the vessels over time. As expected, higher oxygenation was observed in the artery than the vein throughout the experiment. Oxygenation was reduced during the supply of air and increased when 100% $O_2$ was provided. The oxygenation levels in the artery increased faster in response to the change from Air to $O_2$ supply compared to the vein, revealing the expected dynamics of oxygen supply to tissues.

The oxygen extraction rate (OER in Fig. 4c), an indication of the oxygen uptake by cells, was calculated as $OER = (O_{ca} - O_{cv})/O_{ca}$, whereby $O_{ca}$ and $O_{cv}$ is the oxygen saturation in the central artery and vein, respectively[29]. We observed that OER slightly dropped shortly after the Air supply period, suggesting a delayed response in cell oxygenation. When the oxygen supply was increased during the

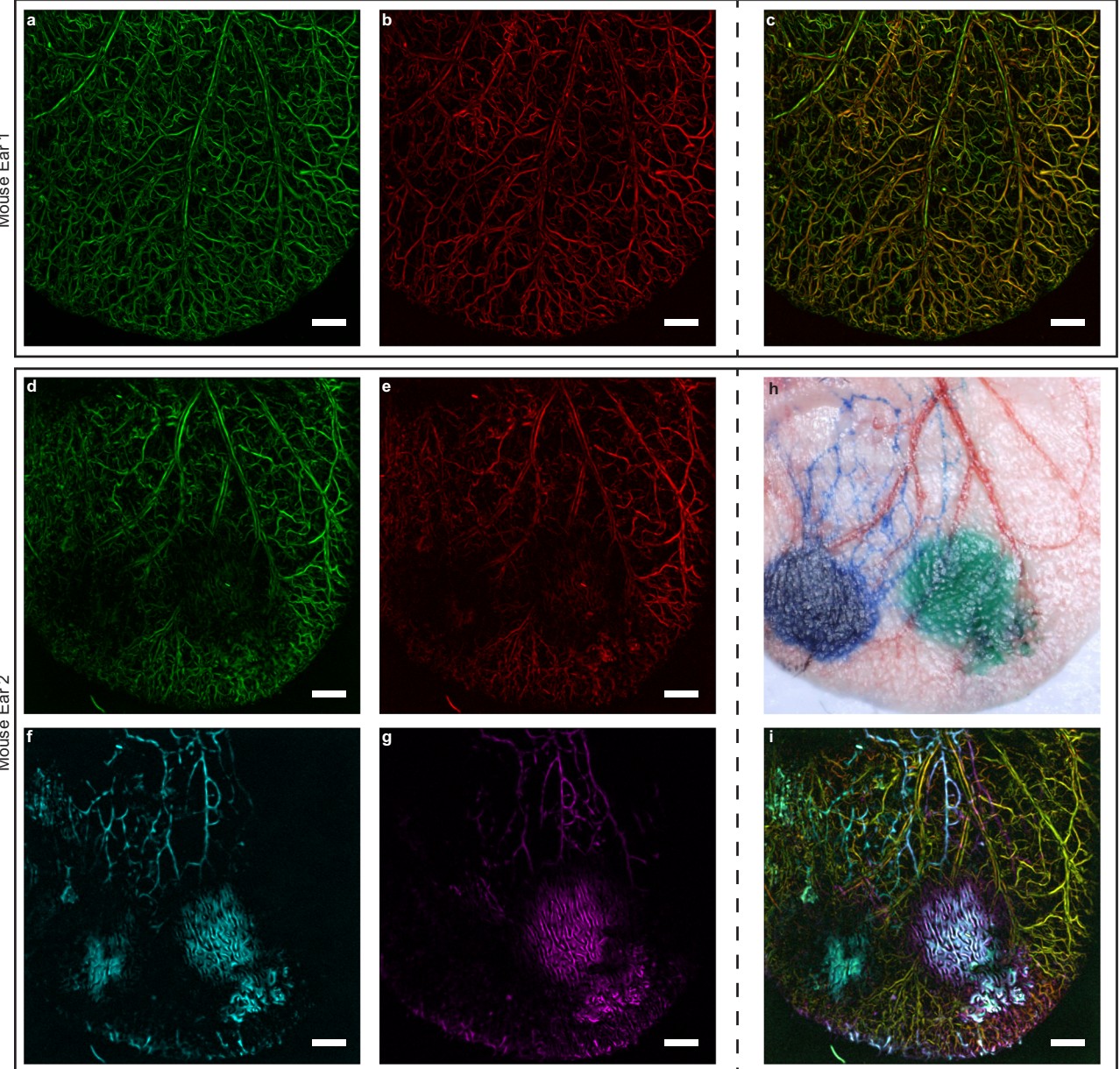

**Fig. 3 | In vivo imaging using FWMOT. a** and **b** A mouse ear at the two blue wavelengths, 445 and 465 nm respectively, with high spatial resolution. **c** The composite image color coded with red indicating higher oxygenation levels compared to green. **d, e, f, g** A second mouse ear at all four wavelengths. **h** A bright-field image of the mouse ear. Intradermal injection spots of Evan's Blue and ICG can be seen in images (**f, g, h**). **f, g** The injected dyes enter the lymphatic vessels that present a different structure than blood vessels. **i** The composite image of all four wavelengths. We can observe oxygenated (red), de-oxygenated (green) blood vessels and lymphatic vessels after uptake of Evan's Blue (cyan) and ICG (purple) at the same time. These experiments were repeated ten times independently with similar results. All images are maximum amplitude projections of reconstructed images. Green is 445 nm, red is 465 nm, cyan is 638 nm, purple is 808 nm, scale bar 1 mm.

second and third $O_2$ supply periods, an increased OER was observed followed by a return to normal levels, suggesting that cells consumed more oxygen as it became available. The oxygen stress test experiment confirms FWMOT as a method that can be employed in studying tissue dynamics with high localization ability, confirmed by intravenous injection of ICG and Evan's Blue. Using the same 2 mm observation field, FWMOT recorded contrast agent dynamics at 638 and 808 nm (Fig. 4d). We observed a similar post-injection pattern for both dyes, revealing two distinctive peaks before settling to a baseline value, indicative of the circulation dynamics of the dyes in the vascular system. We further resolved the delayed appearance of the agents in the vein for both dyes, whereby the observed signal increased at a lower rate, compared to the

artery, a pattern attributed to dye diffusion in the tissue capillary network.

## Discussion
We introduce an optoacoustic method based on frequency multiplexing, which challenges the prevailing notion that TD implementations offer the best optoacoustic performance. By combining the advantages of TD excitation, i.e. pulsed excitation, with FD analysis, FWMOT offers superior performance to TD or FD at multi-wavelength excitation. Increasing the number of wavelengths ($N$) in TD compromises the depth-of-view, SNR, or the total acquisition time. FWMOT can provide simultaneous illumination at multiple wavelengths without sacrificing any of these parameters, enabling either a higher SNR

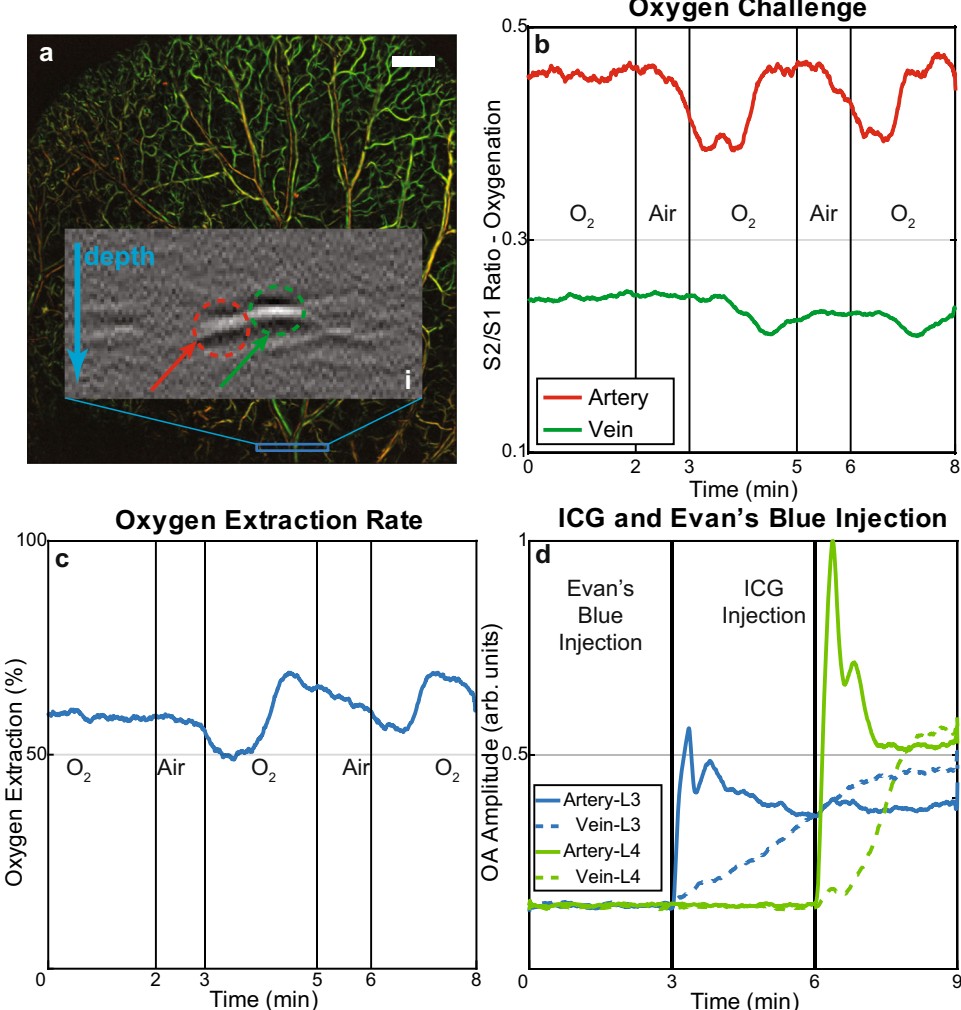

**Fig. 4 | In vivo vascular dynamics revealed by FWMOT. Oxygen challenge experiment and dye injection monitoring in the central vein and artery in the mouse ear using FWMOT imaging. a** The OA images in the blue wavelengths, showing oxygenated (red) and de-oxygenated (green) vessels. We performed continuously B-Scans on the blue region denoted in **a** and 'i' shows such a cross-section. The green arrow and region in 'i' indicate the selected vein, and the red arrow and region indicate the selected artery. Green is 445 nm, red is 465 nm, scale bar 1 mm. **b** The changes in the ratio between the OA signal intensity in wavelength 2 ($S_2$) to that of wavelength 1 ($S_1$) in time during an oxygen stress test. The oxygen saturation is proportional to the ratio $S_2/S_1$. The oxygen saturation changes faster in the artery than in the vein, as expected. **c** The oxygen extraction rate during the same experiment. **d** The signal intensity at wavelengths 3 and 4 ($S_3$ and $S_4$ respectively) at the same artery and vein indicated in **a** during intravascular injection of the two dyes, Evan's Blue and ICG. In both cases the signal intensity increases first in the artery and later in the vein. These experiments were repeated three time independently with similar results.

by a factor of $\sqrt{N}$ per wavelength, shorter acquisition time by a factor of $N$, or $N$ times more depth-of-view per wavelength compared to TDOA. Moreover, the use of pulse trains leads to practical FWMOT implementations by providing the necessary discrete frequencies in the frequency domain, while avoiding generation of FD signals by direct operation in the frequency domain using elaborate multi-frequency modulation systems.

Another critical aspect herein was not only to show the theoretical superiority of the method, but also to demonstrate that it can lead to practical implementations. An enabling technology that allowed such performance was the use of overdriven continuous wave laser diodes (CW-LD)[26] which are inherently cost-effective, portable, and compact, leading to systems with the potential of high dissemination. LDs are ideally suited for FWMOT and can be used to replace solid-state lasers. Besides their small form factor and availability at multiple wavelengths, LDs can be pulsed at very high pulse repetition rates, matching the uniquely optimal FWMOT ability to multiplex and average signals, increasing the SNR. New and more powerful LDs entering the market will contribute to

further improvements of FWMOT performance, offering compact and low-cost systems with high dissemination potential for various applications[9,27].

Imaging at four wavelengths differentiated blood vasculature from lymphatic vessels in vivo, reducing the acquisition time from 2 h required in TD to ~30 min. FWMOT also achieved a fourfold higher acquisition rate compared to TD when performing B-Scans over the main artery and vein, enabling the monitoring of relative oxygenation changes and calculation of oxygen extraction rate. Therefore, FWMOT is particularly suited for dynamic measurements or for experiments that require anesthesia, improving also the associated throughput rate over TD implementations.

There is also a limit to the maximum number of wavelengths used in FWMOT that depends on the UST detection bandwidth, the acquisition time, or the number of averages in the pulse trains and the repetition rate of L1 used in each case. When $f_{rep,1} = 200$ kHz, $N_p = 100$, and the UST bandwidth between 22 and 78 MHz, the maximum number of wavelengths is 28 for FWMOT compared to just 5 for TD implementations with the same operating parameters and a *DoV* of 1.5

mm per wavelength (see Supplementary Note 4). The more wavelengths used in FWMOT, the greater the SNR increase per wavelength compared to TDOA. The use of many simultaneous wavelengths would be particularly appealing for improving the molecular detection specificity of OA spectroscopy.

In summary, FWMOT enables fast, high SNR imaging using multiple wavelengths simultaneously without compromising the DoV and can offer a valuable tool for studying dynamic molecular processes, revolutionizing how multispectral OA imaging will be performed in the future.

## Methods

### 4 Laser diode raster scanning optoacoustic tomography system for FWMOT

Figure 5 presents the multispectral raster scanning optoacoustic mesoscopy (RSOM) system developed to test the advantages of FWMOT. Matlab (Matlab 2016b, Mathworks, USA) was installed on a PC, controlling the system (Fig. 5a). The PC controls the dual-channel stages driver (C-867.260, Physik Instrumente, Germany) that drives the dual $x$–$y$ stages placed on the Scanning Head (Fig. 5c) and is synchronized with two Arbitrary Waveform Generators (AWG, 33522B, Keysight, USA). The AWGs trigger the laser diode drivers and provide a synchronization pulse to the Data Acquisition Card (DAQ, 12-bit, 200MS/s, Razor Express 14x2 Compuscope, Dynamic Signals LLC, USA) for synchronous acquisition of the signal. The light output of the Illumination system is directed into the Scanning Head, with a small portion entering a photodiode (DET10A2/M, Thorlabs, USA). The signal from the photodiode is analog filtered (BLP-90+, Minicircuits, USA) and recorded to monitor the pulse to pulse energy fluctuations and time jitter, corrected for both during signal post-processing. The OA signal from the UST is amplified with a 60dB gain amplifier (Miteq AU-1291-R, Miteq, USA) and analog filtered (BLP-90+ and ZFHP-1R2-S+, Minicircuits, USA) before being digitized by the DAQ to avoid aliasing.

The Illumination system (Fig. 5b) consists of four CW laser diodes that were overdriven with four high-current, short-pulse laser diode drivers developed previously[26]. In short, the peak current of the LDs is briefly (for nanoseconds) increased to >40-fold their CW absolute maximum, which allows the LDs to provide up to 27-fold higher peak power than the manufacturer specified absolute maximum limit. The laser diodes used in this work are the LDM-445-6000 (LaserTack, Germany) emitting at 445 nm, the LDM-465-3500 (LaserTack, Germany) emitting at 465 nm, the HL63283HG (Ushio, Japan) emitting at 638 nm and the K808D02FN (BWT, China) emitting at 808 nm, named laser 1, laser 2, laser 3 and laser 4 respectively. Each laser diode is focused in a multimode fiber. In order to position each laser diode in a manual X–Y stage (CXY1, Thorlabs, USA), a collimating lens (C340TMD, Thorlabs, USA) is placed in front of it on a manual z-stage (SM1Z, Thorlabs, USA), followed by a focusing lens (C560TME, Thorlabs, USA) that is kept stable and the fiber on a x–y stage (CXY1, Thorlabs, USA). The fiber with a 200um core diameter and 0.22NA was one of the 4 inputs of 4x4 fiber power combiner. The four outputs of the fiber combiner (MPC-4-M21-M41-P23, Lasfiberio, China) contain ~25% of the input power of each input fiber and are also multimode fibers with a 200 μm fiber core and 0.22NA. One of the outputs is connected to a custom made 95-5% splitter (LTL 500-93310-95-1, LaserComponents Germany GmBH, Germany) and the 5% fiber was connected to the photodiode. The 3 outputs of the power combiner and the 95% fiber of the splitter were terminated with 1.25 mm ferrules (SFLC230, Thorlabs, USA) and directed to the Scanning Head.

The Scanning Head (Fig. 5c) consists of the x–y stage (U-723 XY, Physik Instrumente, Germany), the 3D printed holder, the ultrasound transducer (UST, HFM23, Sonaxis, France) with a 50 MHz

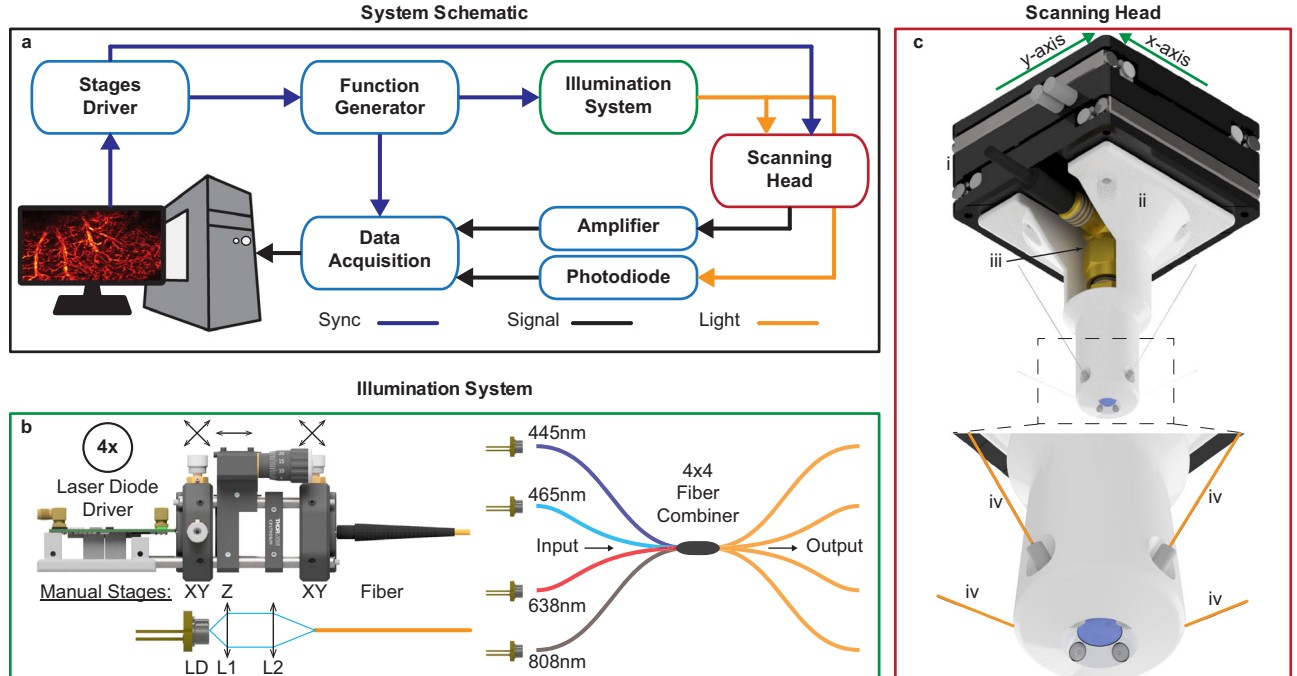

**Fig. 5 | Multispectral Raster Scanning Optoacoustic Mesoscopy system using four laser diodes. a** The schematic of the system developed showing the electrical and optical connections of the different parts of the system. **b** The laser diode illumination system. Each laser diode is attached to a separate laser diode driver and focused into a multimode fiber with a 2-lens system. The four laser diodes are coupled in a 4 × 4 fiber power combiner and each output of the combiner has ~25% of the power of each input, combining all the wavelengths. **c** The scanning head of the RSOM system consisting of the $x$–$y$ scanning stages (i), the 3D printed holder (ii), the Ultrasound Transducer (iii) and the four output fibers of the fiber power combiner (iv) arranged in a circular pattern around the UST. Image components adapted with permission from the copyright holders; the optomechanical components in **b** are retrieved from Thorlabs Inc. and the scanning stages (U-723 XY) in **c** from Physik Instrumente GmbH (https://www.physikinstrumente.com/en/products/xy-stages/u-723-piline-xy-stage-1000583/#downloads).

**Table 1 | Summary of laser diode pulsing and emission characteristics in Frequency Multiplexed Optoacoustic Tomography operation**

| LD | Product name | Provider | Wavelength (nm) | Pulse width (ns) | Repetition rate (Hz) | Energy per pulse (nJ) |
|----|-------------|----------|----------------|-----------------|---------------------|----------------------|
| L1 | LDM-445-6000 | LaserTack | 444.3 ± 1.6 | 6.7 | 200,000 | 189 |
| L2 | LDM-465-3500 | LaserTack | 460.1 ± 1.7 | 6.7 | 200,125 | 137 |
| L3 | HL63283HG | Ushio | 636.8 ± 1.9 | 10.2 | 200,250 | 142 |
| L4 | K808D02FN | BWT | 804.9 ± 2.2 | 10.2 | 200,375 | 153 |

The table presents a summary of the laser diode pulsing and emission characteristics that were used in the frequency wavelength multiplexed optoacoustic tomography setup. For each laser diode the product name, the supplier, the emitted wavelength, the pulse width, the repetition rate and the energy per pulse are presented.

central frequency and 112% relative bandwidth, 3 mm focal length and 0.5 NA, and the four output fibers arranged in a circular pattern around the UST. The output of the four fibers is designed to cross at the focal spot of the UST to achieve a maximum energy density on the sample.

Scanning and recording are performed in a sweeping-like motion using the stages driver as the master in the system. The x-stage moves in a straight line with constant velocity. When the x-stage has traveled a specific distance, equal to the step size, the stages driver sends a signal to the AWGs to trigger the laser diodes. Each laser diode is triggered using a pulse train with repetition rate $f_{rep,1}-f_{rep,4}$ for each laser diode and $N_{p1}-N_{p4}$ number of pulses at each point (A-Scan) in the B-Scan. After the stage has traveled the desired distance, the B-Scan is completed and the stages stop moving. The y stage is moved to the next y-position and the x-stage can now perform the next B-Scan in the opposite direction. For the FWMOT excitation we used $f_{rep,1} = 200,000$ Hz, $f_{rep,2} = 200,125$ Hz, $f_{rep,3} = 200,250$ Hz and $f_{rep,4} = 200,375$ Hz and $N_{p1} = 100$ and $N_{p2} = N_{p3} = N_{p4} = 101$ pulses for lasers 1, 2, 3, and 4, respectively. For a detailed derivation of these values, see Supplementary Note 2.

We recorded the emission spectra of the 4 laser diodes used in the multispectral laser diode RSOM with a spectrometer (USB4000, OceanOptics, UK) and the peak emission wavelengths were 444.3, 460.1, 636.8, 804.9 nm with a variance of 1.6, 1.7, 1.9, 2.2 nm respectively and an $R^2$ confidence level higher than 0.96 for all cases.

Additionally, the energy per pulse on samples was measured with a stabilized thermal power meter (PM160T, Thorlabs, USA) and calculated as 189, 137, 142, 153 nJ per pulse for lasers 1, 2, 3, 4, respectively. The pulse width was estimated as 6.7, 6.7, 10.2, 10.2 ns full-width-at-half-maximum (FWHM) for each laser respectively. Using a USB CCD camera (daA1920-30 µm; Basler AG, Germany) the illumination spot on the surface of the sample was measured to be a circle with a diameter of ~1 mm. Table 1 presents a summary of the LD pulsing and emission characteristics for FWMOT operation as presented above.

To compare the FD OA to FWMOT excitation we used a fiber-coupled 450-nm laser diode (FBLD-450-0.8W-FC105-BTF; Frankfurt Laser Company, Germany) connected to an analog laser driver (BFS-VRM 03 HP; Picolas, Germany). The output fiber of the laser was pumped into one of the input fibers of the 4 × 4 fiber combiner, so that the illumination of the sample is identical for the FD excitation and the FWMOT excitation system.

To calculate the Signal-to-Noise ratio (SNR) we used the following formula, $10 * \log_{10}(S / N)$, where $S$ is the signal intensity and $N$ the standard deviation of the noise floor.

## Signal processing, reconstruction algorithm, and system resolution

Image acquisition occurs in a large Field of View ($10 \times 10$ mm$^2$) and with a scanning step size of 10 µm in order to be much lower than the lateral resolution of the system. Time-resolved signals detected at each scanning position correspond to the integration of acoustic spherical waves originating from the illuminated optical absorbers within the detection angle of the transducer. Therefore, unprocessed images obtained directly from the system are heavily blurred, and require further processing to obtain high-contrast and high-resolution images. To do so, a back-projection algorithm is implemented in the Fourier domain[30,31] to recover an (acoustically) diffraction-limited image. The resulting image is then corrected by the system impulse response[32] and further processed with a vesselness filter for display purposes[33]. The raw data of a single B-Scan do not allow for 3D image reconstruction. Therefore, a simplified version of the back-projection algorithm was developed, which operates in two dimensions and approximates the transducers sensitivity field as a conic section along the B-Scan direction only.

To calculate the system's spatial resolution we imaged a resolution target (a Siemen's star) using all 4 wavelengths. The results are presented in Supplementary Fig. 3, which shows the reconstructed OA images of the resolution target at all four wavelengths, as well as the composite image in polar coordinates. The spatial resolution is given by the smallest radius at which all the lines can be well resolved (Supplementary Fig. 3h), which is 38 µm for all wavelengths.

## Laser diode emission spectra, hemoglobin and dye absorption spectra

Hemoglobin has a broad absorption spectrum over the visible and near-infrared range with a higher absorption at the lower wavelengths of the spectrum. The absorption of de-oxygenated hemoglobin is higher than the absorption of oxygenated hemoglobin at 444 nm. The absorption of oxygenated hemoglobin is higher than the absorption of de-oxygenated hemoglobin at 460 nm. The total absorption of hemoglobin at wavelengths 1 and 2 is much higher than that at wavelengths 3 and 4. Due to the low energy output of the laser diodes, this is enough to assume that we only detected OA signals from hemoglobin at wavelengths 1 and 2. This has been confirmed from the in vivo mouse ear experiments. Moreover, we can estimate the relative changes of oxygen saturation[34] by calculating the ratio of the signal intensity of wavelength 2 over that of wavelength 1, $S_2/S_1$, with a higher ratio indicating a higher oxygen saturation.

To induce contrast and increase the SNR at wavelengths 3 and 4 we used two dyes, Evan's Blue (Sigma-Aldrich, Germany) and ICG (VERDYE, Germany). Evan's Blue has a peak absorption at 640 nm and a minimum at 740 nm[35] making it the appropriate dye to enhance the contrast for wavelength 3, since this is the only wavelength where we can detect OA signals. ICG in blood plasma has a peak absorption at around 810 nm and a low absorption at 637 nm[36], making it appropriate to induce OA contrast at wavelength 4 with minimal contrast at wavelength 3 and no contrast at the other wavelengths, confirmed by the mouse ear experiments.

## Maximum permissible exposure (MPE) limits compliance and presence of non-linear effects

Using the above-mentioned values (Table 1) for each laser diode and a scanning speed of 10 mm/s for a scanning step size of 10 µm we can calculate the sample exposure. The total exposure of the sample for simultaneous illumination with all four wavelengths is calculated to be 19.8 µJ/cm$^2$ per pulse and 3.96 W/cm$^2$ mean exposure, well below the MPE limits of 20 mJ/cm$^2$ and 18 W/cm$^2$, imposed by the American

National Standards Institute[37]. It has been demonstrated that non-linear effects in OA are present at high energy densities above ~7 mJ/cm$^2$ [38]. The total sample exposure herein was less than 20 $\mu$J/cm$^2$, even in the case of simultaneous illumination with all wavelengths. Therefore, the deposited energy in this study was much lower than the threshold for non-linear effects.

## Mouse handling and imaging protocol

For our experiments, we employed two 5- to 6-week-old female Athymic nude- Foxn1$^{nu}$ mice (Envigo, Germany). During all measurements, the mice were anesthetized by 1.6% Isoflurane (cp-Pharma, Germany) with 0.8 lpm carrier gas flow and body temperature was maintained with an infrared heat lamp and heating plate.

The first mouse was used for the experiments presented in Fig. 3. The lymphatic ear vessels were highlighted by intradermal administration of 5 $\mu$l ICG (5 mg/ml) and 5 $\mu$l Evan's blue (1%) into the ear tip of the mouse. ICG was administered ~30 min and Evan's blue ~10 min before starting imaging to ensure lymphatic drainage of the dyes.

A second mouse was used for the experiments demonstrated in Fig. 4. For the oxygen stress experiment, we supplied different isoflurane carrier gas combinations or breathing conditions through a nose mask. The mouse was breathing alternatively 0.8 lpm of 100% oxygen ($O_2$) and a combination of 0.6 lpm medical air (20% Oxygen) plus 0.2 lpm carbon dioxide $CO_2$ (Air). For the dye diffusion experiment we first acquired the background data and after 3 min, intravenously injected 100 $\mu$l of 1% Evan's Blue solution, and after another 3 min, 100 $\mu$l of the 5 mg/ml ICG solution. Both mice were sacrificed immediately after imaging by cervical dislocation.

All mouse experiments were performed according to the committee on Animal Health and Care of Upper Bavaria, Germany (Az. ROB-55.2-2532.Vet_02-14-88 and Az. ROB-55.2-2532.Vet_02-18-120). The mice were maintained in an individual ventilated cage system (Tecniplast, Germany) at 22° ambient temperature, ~50% relative humidity, and with a regular 12 h day/night cycle in our specific-pathogen-free (SPF) mouse facility at the Center for Translational Cancer Research of the Technical University of Munich (TranslaTUM).

## Human skin experiment

One of the authors volunteered for this study to have their hand imaged. After consultation with the TUM Ethics Commission, no formal ethics approval was necessary. Informed consent from the participant was obtained and archived. The participant is a 30-year-old white male. Image acquisition occurred in a Field of View of 5 × 5 mm$^2$ and with a scanning step size of 10 $\mu$m. The reconstructed OA images are presented in Supplementary Fig. 2.

## Reporting summary

Further information on research design is available in the Nature Research Reporting Summary linked to this article.

# Data availability

The raw optoacoustic signal data for validating the frequency wavelength multiplexed optoacoustic tomography algorithm, the raw imaging data of mice and humans, the raw B-Scan data of the Oxygen stress test and ICG and Evan's Blue injection experiments that were generated in this study have been deposited in the Zenodo database under accession code: https://doi.org/10.5281/zenodo.6770729.

# Code availability

The authors declare that for data collection the commercially available software from GaGe (Compuscope Driver Version 5.04.34, Dynamic Signals LLC, USA) and Matlab 2016b (Matlab, Mathworks, USA) was used. Data analysis was conducted in Matlab using its built-in functions. The frequency wavelength multiplexed optoacoustic tomography algorithm has been patented and is available upon discretion from the corresponding author.

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

## Acknowledgements

The research leading to these results has received funding by the Bundesministerium für Bildung und Forschung (BMBF), Bonn, Germany (Project Sense4Life, 13N13855) (V.N.), from the European Research Council (ERC) under the European Union's Horizon 2020 research and innovation program under grant agreement No 694968 (PREMSOT) (V.N.) and from the European Union's Horizon 2020 research and innovation program under grant agreement No 732720 (ESOTRAC) (V.N.) and No 862811 (RSENSE) (V.N.). We would like to thank Dr. Sergey Sulima and Dr. Robert Wilson for their help writing the manuscript.

## Author contributions

A.S. and L.P. developed the multiple wavelength RSOM system. L.P. developed the Frequency Wavelength Multiplexed signal processing technique. AS further worked on the Frequency Wavelength Multiplexed algorithm, developed the overdriven CW-LD excitation source, performed the imaging experiments and processed the data. SG assisted AS in performing the mice experiments. Q.M. reconstructed the optoacoustic images. C.Z. and V.N. helped with extensive discussions and guidance in conducting this research.

## Funding

## Competing interests

V.N. is a founder and equity owner of sThesis GmbH, iThera Medical GmbH, Spear UG, and i3 Inc. The remaining authors have no competing interests.
