## [Peer Review File · Nature Communications]

Frequency Wavelength Multiplexed Optoacoustic TomographyREVIEWER COMMENTS

Reviewer #1 (Remarks to the Author):

The manuscript entitled “Frequency comb optoacoustic tomography” given by Stylogiannis et al. reports a new method in optoacoustic tomography. The authors demonstrate an efficient multi-wavelength simultaneous excitation technique with four periodic nanosecond pulse trains whose repetition rates differ slightly. The slightly different periodicities work as intensity modulations and allow one to demodulate the time-domain optoacoustic time-of-flight signal. Thanks to the simultaneous excitation and demodulation, one can make 4-wavelengths-excitation time-domain photoacoustic measurements simultaneously, e.g., 4-times faster measurement.

The demonstrated technique is valuable in the field of optoacoustic tomography, but the manuscript is written in a misleading way. In particular, the authors call the method “Frequency comb optoacoustic tomography,” but it is not related to frequency comb technology at all. This technique uses the repetitiveness of the nanosecond lasers as intensity modulations to demodulate the signal in the Fourier domain. It is quite a typical concept that we can explain apart from the frequency comb. One should use the term “frequency comb” only when it is used the laser’s temporal coherence, including high phase stability. I request the authors rewrite the manuscript in a way that does not use the term frequency comb.

Another misleading point is the definition of the time- and frequency-domain. I think the authors' technique is not a frequency-domain technique but a multiplex time-domain technique because it still uses the time-of-flight modality. As the authors mention in the introduction, for example, OCT historically moved from time-domain to frequency-domain, but the latter benefits by directly measuring the spectrum. The authors' technique takes temporal waveform and Fourier-transform it to get a spectrum, which is similar to the time-domain OCT. I request the authors re-consider this point and make a proper modification.

The other comments are listed below.

1. A conceptual explanation about photoacoustic measurement is missing in the main text. For example, how the acoustic signal is generated and detected? What is the time scale of the phenomenon? For non-experts, these general explanations should be explained.
2. The temporally multiplex excitation would cause spurious artifacts due to the nonlinear effects, detector saturation, or sample damage when the pulses from different lasers overlap, which one cannot avoid. The authors must discuss this drawback carefully.

3. All the figures that explain the mechanism and the related descriptions are made based on the delta-function-like spectrum in the frequency domain. However, there is linewidth determined by the measurement time. Since this is an essential factor to understand this technique correctly, it should be discussed more in the main text.
4. The figures of experimental data about the temporal waveforms and its Fourier-transformed spectra of the FCOT measurements are missing, although they are the main significance of this manuscript. Therefore, one cannot verify the heart of the technique. I request the authors to make additional figures that show them.
5. Multimode fibers are used for delivering the excitation light. I would expect the speckle effect due to the fiber. How do the authors mitigate the speckle effect?
6. Spatial resolution is not discussed except in the section "Reconstruction algorithm" of Methods. Since this is a paper about an imaging technique, the spatial resolution should be discussed in the main text.
7. In the methods, the specifications of the laser diodes are described without a figure. It would be helpful to show the data as a figure, perhaps in the supplementary information, to verify it.

Reviewer #2 (Remarks to the Author):

This is a well-written manuscript with a valuable new approach in Optoacoustic Tomography demonstrating good results; it deserves publication in Nature Communications. The approach and method are explained in the manuscript in detail.

However, I am not a specialist in the field and I had to go through the manuscript in detail, to understand the importance of the results. My main criticism is that the introduction needs improvement. The field and the importance of the new method to the field needs a good introduction, not only referring to other publications. For example, line 40-49 are without further explanation and are only quickly understandable for a specialist.

I also recommend reconsidering the abstract. The emphasis in the first lines on the differences between the Time domain and the Frequency domain is a very technical comment.

In addition, the Discussion also needs a proper outlook into the field. How can the new method influence the research and bring new opportunities to the field; this is not convincingly written.

The title mentions Frequency Combs. As such, I would expect more proper introduction of FCs to this field.

Other comments:

- Line 64: I am not familiar with the expression: “overdriving laser diodes”, please explain.
- Lines 127: I assume this should be Fig.2b?
- Line 259 FCOT can produce depth resolve images (see Suppl FigS2). However, I cannot find these there. Unless this should be FigS3. I propose to use in these images a vertical scale (1 mm?)
- Line 419,420, 468 etc. use μ instead of u
- FigS1c, details of the scanning head are difficult to see on paper, I propose to enlarge it.

Reviewer #3 (Remarks to the Author):

This paper presents two very compelling advancements in optoacoustic tomography (OT). The investigators developed a new (to OT) frequency comb paradigm that mixes both frequency domain and time domain approaches to encoding illumination and decoding the response. The primary use of this is to encode different optical wavelengths to enable better spectroscopy. The critical performance advantage is an improvement in SNR that scales as the square root of the number of optical wavelengths employed. This encoding scheme is coupled with an advancement in replacing solid-state lasers with cheaper pulsed laser diodes. There is an interaction between two ideas and combined together, the laser diode FCOT paradigm provides a practical system cable of high-performance in-vivo imaging. The concept is particularly appealing, as they are borrowing well-developed frequency comb strategies from other disciplines and introducing them to the field of Optoacoustic Tomography, while the laser diodes provide an instrument development strategy that can scale the wavelengths. However, to establish the improvement of FCOT the SNR analysis should involve a more extensive quantitative evaluation of the measurements.

Major critiques:

1. While the improvements over of FCOT over time-domain OT can be evaluated through either comparative frame rates or comparative SNR, since the faster imaging is a chosen setting, the performance goal of SNR is the more important feature to validate. While this nominally done in Figure 2 (e.g. 2h), where it is stated that “the SNR of the optical acoustic signal is reduced (for traditional OT)” the results are not clear. While from an \sqrt{N} analytical perspective a 2x increase in SNR would be

expected - it is not clear that this was measured in a quantitative analysis of the data. This should be done, and it is likely fairly straightforward.

2. This comment is an extension of the first. Which feature in figure 2 is demonstrating the improved SNR? What should one look at? Maybe a plot of the residuals would show it better? Not sure. Possibly the feature is the background variance vs the "signal" structure. To these untrained eyes, plots 2f-l all look very similar, with no discernable difference between the methods. Because SNR is the main point of the paper, the visual intuition of the improvement should be much clearer and cleaner.

3. SNR analysis is not presented in either figure 3 or 4. For example, Figure 3 is impressive, with very detailed imaging of the ear. But there is no analysis of SNR. Since the main point of the FCOT is SNR, it would seem logical to have an SNR comparison with the in-vivo data also. That also is likely straightforward to implement, possibly with an alternate analysis of the existing data. This analysis is critical, otherwise Figs. 3 and 4 have limited verification/validation value.

4. The discussion of extending up to 28 wavelengths is intriguing because the SNR improvements become more dramatic with increasing n . Since this topic is central to the paper this analysis and prospect should be further developed. In what scenarios would 28 wavelengths be useful? Are there limits to how many frequencies can be encoded?

Minor Comments:

1. It is easy to accept that all of MRI is in the frequency domain. But is it true that all of OCT is in the frequency domain? Some OCT is swept-source that probably is a bit more like sequential time domain, a few references with clarification would be helpful.

2. In Figure 2, the labels of c and d, with the $4 \times N_p/4$ and red circles are difficult to follow. The motivation is understandable. Keeping track of things is tricky because of the interaction between acquisition time, the number of pulses, etc... but if there is a way to improve the clarity that would help many readers.

Point-by-Point Responses to Reviewer Comments

We thank the reviewers for thoroughly evaluating our work and providing thoughtful suggestions for improvement. We have endeavored to address as many of the suggestions as possible in our revised manuscript. Below are point-by-point responses to the reviewers' comments. The color code is as follows: *reviewer comments are in blue*, our responses are in black, and *additions or changes to the text are in orange*.

Reviewer 1

The manuscript entitled “Frequency comb optoacoustic tomography” given by Stylogiannis et al. reports a new method in optoacoustic tomography. The authors demonstrate an efficient multi-wavelength simultaneous excitation technique with four periodic nanosecond pulse trains whose repetition rates differ slightly. The slightly different periodicities work as intensity modulations and allow one to demodulate the time-domain optoacoustic time-of-flight signal. Thanks to the simultaneous excitation and demodulation, one can make 4-wavelengths-excitation time-domain photoacoustic measurements simultaneously, e.g., 4-times faster measurement.

The demonstrated technique is valuable in the field of optoacoustic tomography, but the manuscript is written in a misleading way. In particular, the authors call the method “Frequency comb optoacoustic tomography,” but it is not related to frequency comb technology at all. This technique uses the repetitiveness of the nanosecond lasers as intensity modulations to demodulate the signal in the Fourier domain. It is quite a typical concept that we can explain apart from the frequency comb. One should use the term “frequency comb” only when it is used the laser’s temporal coherence, including high phase stability. I request the authors rewrite the manuscript in a way that does not use the term frequency comb.

Another misleading point is the definition of the time- and frequency-domain. I think the authors' technique is not a frequency-domain technique but a multiplex time-domain technique because it still uses the time-of-flight modality. As the authors mention in the introduction, for example, OCT historically moved from time-domain to frequency-domain, but the latter benefits by directly measuring the spectrum. The authors' technique takes temporal waveform and Fourier-transform it to get a spectrum, which is similar to the time-domain OCT. I request the authors re-consider this point and make a proper modification.

We agree with the reviewer that the term Frequency Comb might be confusing for some readers. Therefore, we have changed the terminology to Frequency Multiplexed Optoacoustic Tomography throughout the whole text.

The other comments are listed below.

Comment 1

A conceptual explanation about photoacoustic measurement is missing in the main text. For example, how the acoustic signal is generated and detected? What is the time scale of the phenomenon? For non-experts, these general explanations should be explained.

We would like to thank the reviewer for bringing this into our attention. We have revised the introduction to explain the OA signal generation and detection.

Line 31-40:

Generation of photoacoustic (OA) signals requires illumination of the sample with energy transients (e.g., pulsed or sinusoidal illumination)¹. The sample absorbs this time-variant energy and subsequently generates an acoustic wave through thermo-elastic expansion². Time Domain (TD) OA implementations offer large energy transients by means of nanosecond duration light pulses³⁻⁶, in order to satisfy the thermal and stress confinement limits needed for photoacoustic signal generation⁷. A nanosecond duration pulse also maximizes the energy transient and optimizes the signal-to-noise ratio (SNR), thus making TD the domain of choice in photoacoustics⁸⁻¹⁰. TD photoacoustic imaging records the time-of-flight of the generated ultrasound waves (US) at multiple locations on the surface of the interrogated object by means of a sensitive ultrasound transducer and, using mathematical inversion, converts these measurements to three-dimensional maps of optical absorption¹¹.

Comment 2

The temporally multiplex excitation would cause spurious artifacts due to the nonlinear effects, detector saturation, or sample damage when the pulses from different lasers overlap, which one cannot avoid. The authors must discuss this drawback carefully.

We are aware of the existence of non-linear effect in OA. However, it has been proven that the threshold above which the non-linear effects are present is very high ($\sim 7\text{mJ}/\text{cm}^2$). Even in the case of simultaneous illumination of the sample with all the wavelengths used in this study, the sample exposure is $19.8\ \mu\text{J}/\text{cm}^2$, much lower than the threshold value. Therefore, the non-linear effects are not present in FMOT. We have added this explanation in the Methods section, Lines 520-523:

It has been demonstrated that non-linear effects in OA are present at high energy densities above $\sim 7\text{mJ}/\text{cm}^2$ ³⁸. The total sample exposure herein was less than $20\ \mu\text{J}/\text{cm}^2$, even in the case of simultaneous illumination with all wavelengths. Therefore, the deposited energy in this study was much lower than the threshold for non-linear effects.

Comment 3

All the figures that explain the mechanism and the related descriptions are made based on the delta-function-like spectrum in the frequency domain. However, there is linewidth determined by the measurement time. Since this is an essential factor to understand this technique correctly, it should be discussed more in the main text.

We would like to thank the author for his comment. We have attempted to explicitly explain the limitations of the acquisition time and the frequency resolution, Δf , on the choice of the frequency shift between the different lasers, δf , in the Results section, Lines 193-203. We have also provided a specific derivation of all the formulas in the Supplementary material Lines 56-110. We also discuss the effect it has on the maximum number of wavelengths in FMOT in the Discussion Lines 368-375. We think that the existing text in the manuscript explains adequately the implications the existing linewidth in the frequency domain has.

In addition, we believe that Fig.1i showcases the appearance of a real OA power spectrum for one wavelength. To complement Fig.1i, and show the limitations imposed by the limited acquisition time and the linewidth in the frequency domain we have added two figures to the Supplementary data with the LD excitation time series and spectrum presented in Fig.S4 and the OA signal time series and power spectrum in Fig.S5. These figures also help explain the Frequency Multiplexed OA scheme with multiple wavelengths. We believe that these figures further showcase the effect of the finite linewidth in the frequency domain with real-life data and help clarify the limitations of our method as explained in the manuscript. See next comment for added text.

Comment 4

The figures of experimental data about the temporal waveforms and its Fourier-transformed spectra of the FCOT measurements are missing, although they are the main significance of this manuscript. Therefore, one cannot verify the heart of the technique. I request the authors to make additional figures that show them.

Following this and the previous comment of this reviewer, we added two figures in the Supplementary data as described above (Fig.S4-5), as well as the following text.

Lines 206-208:

Supplementary Figures 4 and 5 present the Laser Diode excitation time series, the corresponding OA signal time series and the power spectra of both, in order to further showcase FMOT's operation and verify the method's ability to recover each signal without cross talk.

Supplementary Material Lines 112 – 144:

Supplementary Figure 4 presents the LD Excitation Time Series for FMOT using 4 wavelengths, with $f_{rep1}=200000\text{Hz}$ and $\delta f=125\text{Hz}$ with $N_{avg}=200$. Figure S4.a presents the full time series, which displays how each wavelength has a slightly different repetition rate. Fig S4.b presents the 4 first

pulses of the pulse train. The LD pulses of L1 and L2 are very close to each other and cannot be separated at this early stage. The same is true for the LD pulses of L3 and L4, only that there is time delay between L1-L2 and L3-L4. The delay is there because L1 and L2 are triggered by the same AWG (AWG1), which is triggered by the stage or the PC. However, L3 and L4 are triggered by a second AWG (AWG2), which is triggered by AWG1, introducing a small time shift. Fig S3.c shows the 16th pulse of each pulse train. The pulses of each distinct LD are now visible. Fig S3.d show the last 4 pulses of each pulse train, where slightly different periods of each LD are apparent.

Fig S4.e demonstrates the power spectrum of the LD pulse train (Fig S4.a) between 0-100 MHz. Here, a small shift between the harmonics of the different lasers can be observed. Fig S4.f present the power spectrum at the lower end of the UST bandwidth, between 14-15 MHz; the harmonics of each LD can be distinguished. Fig S4.g shows the power spectrum near the center of the UST bandwidth, between 32-36 MHz and Fig S4.h at the upper end of the UST bandwidth, between 85-86 MHz, where the harmonics of each LD are spread further apart, as expected.

We then used the above LD excitation series to generate and detect an OA signal from a black varnish layer on a petri-dish. The resulting time series and power spectrum of the generated OA signal are presented in Fig S5 for the same time points (Fig S5.(a-d) correspond to Fig S4.(a-d)) and frequency range (Fig S5.(e-h) correspond to Fig S4.(e-h)), respectively. We observe that the laser trigger interference appears at the same time points as the laser pulse; the OA signal is expected later in time as it has to travel from the sample to the UST. The OA signal is generated at the excitation frequencies and by choosing the correct set of harmonics for each laser, which can be decomposed to frequency multiplexed signals.

Comment 5

Multimode fibers are used for delivering the excitation light. I would expect the speckle effect due to the fiber. How do the authors mitigate the speckle effect?

We understand the reviewer's concerns; however, the speckle effect is not a challenge in OA in general. One of the main advantages of OA is that the resolution does not depend in the optical diffraction limited spot, for acoustic resolution OA microscopy and OA tomography. Therefore, light is diffused in the interrogated sample and the spatial resolution depends on the ultrasound focusing. As a consequence, the presence of any speckles will be either too weak to generate a detectable OA signal or much smaller than the actual spatial resolution of the system, which is 38 μm in our case.

Comment 6

Spatial resolution is not discussed except in the section "Reconstruction algorithm" of Methods. Since this is a paper about an imaging technique, the spatial resolution should be discussed in the main text.

We would like to thank the reviewer for bringing this point to our attention. We have included in the Supplementary data a figure (Fig. S3) with OA images of a resolution target and the achieved resolution. We also added a short text in the Methods section Lines 487-491:

To calculate the system's spatial resolution we imaged a resolution target (a Siemen's star) using all 4 wavelengths. The results are presented in Supplementary Fig.S3, which shows the reconstructed OA images of the resolution target at all four wavelengths, as well as the composite image in polar coordinates. The spatial resolution is given by the smallest radius at which all the lines can be well resolved (Fig.S3h), which is 38 μm for all wavelengths.

Comment 7

In the methods, the specifications of the laser diodes are described without a figure. It would be helpful to show the data as a figure, perhaps in the supplementary information, to verify it.

In order to clarify the system's design and the LD application, we have now moved the former Fig. S1 (the system Figure) to the main manuscript; the new Fig.5 can now be found in the Methods section. As a consequence, we have simplified the last two paragraphs of the Introduction, Lines 73-82:

*Following theoretical considerations, we hypothesized that overdriven laser diodes²⁶, which offer a cheaper and more practical alternative to solid state lasers, could exploit FM pulse trains and lead to high quality optoacoustic imaging that could demonstrate benefits over TD implementations. We introduce a **Frequency Multiplexed Optoacoustic Tomography (FMOT)** system that utilizes 4 concurrently pulsing overdriven laser diodes, each utilizing a slightly different repetition rate in order to encode different wavelengths. These wavelengths appear then at different discrete frequencies in the frequency domain. We show concurrent multi-wavelength mesoscopic²⁷ imaging of lymphatic and microvascular dynamics in mice at high SNRs, offering the fastest multi-wavelength illumination ever achieved in the field of optoacoustics and confirming spectral performance that improves upon TD implementations.*

Additionally, in order to present the laser diode pulsing and emission characteristics in a more comprehensible manner, we have restructured the Methods section to group all of the LD characteristics in the same subsection (**4 Laser Diode Raster Scanning Optoacoustic Tomography System for FCOT**) in Methods and included Table 1 to serve as a summary of all the values characterizing the laser diode emission in Lines 458-462:

The following table presents a summary of the LD pulsing and emission characteristics for FMOT operation as presented above.

Table 1 Summary of Laser Diode pulsing and emission characteristics in Frequency Multiplexed Optoacoustic Tomography operation.

LD	Product Name	Provider	Wavelength (nm)	Pulse Width (ns)	Repetition Rate (Hz)	Energy per pulse (nJ)
L1	LDM-445-6000	LaserTack	443.3 ± 1.6	6.7	200 000	189
L2	LDM-465-3500	LaserTack	430.1 ± 1.7	6.7	200 125	137
L3	HL63283HG	Ushio	636.8 ± 1.9	10.2	200 250	142
L4	K808D02FN	BWT	804.9 ± 2.2	10.2	200 375	153

We also believe that Supplementary Figures S4 and S5 improve the reader's understanding of the LD specifications in regard to the FMOT multiplexing scheme.

Reviewer 2

This is a well-written manuscript with a valuable new approach in Optoacoustic Tomography demonstrating good results; it deserves publication In Nature Communications. The approach and method are explained in the manuscript in detail.

However, I am not a specialist in the field and I had to go through the manuscript in detail, to understand the importance of the results. My main criticism is that the introduction needs improvement. The field and the importance of the new method to the field needs a good introduction, not only referring to other publications. For example, line 40-49 are without further explanation and are only quickly understandable for a specialist.

We would like to thank the reviewer for the comment. We have taken it into account by revising the Introduction to improve its comprehensibility for non-experts.

Lines 44-58:

Frequency Domain (FD) optoacoustics has been also considered as an alternative to TD, by modulating the illumination intensity at a discrete frequency and detecting the generated OA signals at the same frequency¹⁴⁻¹⁶. Signal detection is achieved with demodulation techniques that retrieve the amplitude and phase of the OA signal, a technology that is simpler and more economic than recording time signals at tens of MHz sampling rates, as is common in TD detection. FD can also enable concurrent illumination at multiple wavelengths, by modulating sources of different colour at different frequencies¹⁷⁻¹⁹. Despite these advantages, intensity-modulated light¹⁴⁻¹⁶ provides energy transients and corresponding optoacoustic signals that are as low as six orders of magnitude¹⁹ weaker than the ultrashort pulses used in TD, drastically

reducing the SNR in the FD²⁰⁻²². Moreover, optoacoustic investigations at a single frequency fail to collect depth information or lead to three-dimensional imaging. We have recently shown²³ that depth information and three-dimensional image reconstruction requires the generation of signals at multiple discrete frequencies, a requirement that leads to complex emission (modulation) and detection (demodulation) schemes^{23,24}. Therefore, despite the potential advantages over TD^{17-19,23,24}, FD has had little impact in the field of optoacoustics.

I also recommend reconsidering the abstract. The emphasis in the first lines on the differences between the Time domain and the Frequency domain is a very technical comment.

We have attempted to improve the phrasing in the abstract to (briefly) highlight the advantages and disadvantages of TD and FD optoacoustics.

Lines 18 – 28:

Optoacoustics (OA) is overwhelmingly implemented in the Time Domain (TD) to achieve high signal-to-noise ratios. Implementations in the Frequency Domain (FD) have been proposed, but suffer from low signal-to-noise ratios and have not offered competitive advantages over time domain methods to reach high dissemination. It is therefore commonly believed that TD is the optimal way to perform optoacoustics. We introduce a novel optoacoustic concept based on pulse train illumination and frequency domain multiplexing and theoretically demonstrate the superior merits of the approach compared to the time domain. Then, using recent advances in laser diode illumination, we launch Frequency Multiplexing Optoacoustic Tomography (FMOT), at multiple wavelengths, and experimentally showcase how FMOT optimizes the signal-to-noise ratios of spectral measurements over time-domain methods in phantoms and in vivo. We further find that FMOT offers the fastest multi-spectral operation ever demonstrated in optoacoustics.

In addition, the Discussion also needs a proper outlook into the field. How can the new method influence the research and bring new opportunities to the field; this is not convincingly written.

We have made changes and additions to the discussion to more clearly state the outlook for FD optoacoustics (also in response to comments by the other reviewers).

Lines 338-359:

We introduce an optoacoustic method based on frequency multiplexing, which challenges the prevailing notion that TD implementations offer the best optoacoustic performance. By combining the advantages of TD excitation, i.e. pulsed excitation, with FD analysis, FMOT offers superior performance to TD or FD at multi-wavelength excitation. Increasing the number of wavelengths (N) in TD compromises the depth of view, SNR or the total acquisition time. FMOT can provide simultaneous illumination at multiple wavelengths without sacrificing any of these parameters, enabling either a higher SNR by a factor of \sqrt{N} per wavelength, shorter acquisition time by a factor of N, or N times more depth of view per wavelength compared to

TDOA. Moreover, the use of pulse trains leads to practical FMOT implementations by providing the necessary discrete frequencies in the frequency domain, while avoiding generation of FD signals by direct operation in the frequency domain using elaborate multi-frequency modulation systems.

Another critical aspect herein was not only to show the theoretical superiority of the method, but also to demonstrate that it can lead to practical implementations. An enabling technology that allowed such performance was the use of overdriven continuous wave laser diodes (CW-LD)²⁶ which are inherently cost-effective, portable and compact, leading to systems with the potential of high dissemination. LDs are ideally suited for FMOT and can be used to replace solid-state lasers. Besides their small form factor and availability at multiple wavelengths, LDs can be pulsed at very high pulse repetition rates, matching the uniquely optimal FMOT ability to multiplex and average signals, increasing the SNR. New and more powerful LDs entering the market will contribute to further improvements of FMOT performance, offering compact and low-cost systems with high dissemination potential for various applications^{9,27}.

The title mentions Frequency Combs. As such, I would expect more proper introduction of FCs to this field.

We have now changed the term “Frequency Comb” to “Frequency Multiplexing” in response to a comment from reviewer 1.

Comment 1

Line 64: I am not familiar with the expression: “overdriving laser diodes”, please explain.

The reviewer correctly addresses that fact that it is not adequately explained in the manuscript and we added a short explanation. However, since the overdriving concept is separately published in a previous publication we added a short text in the Methods section Lines 411-414:

In short, the peak current of the LDs is briefly (for nanoseconds) increased to >40-fold their CW absolute maximum, which allows the LDs to provide up to 27-fold higher peak power than the manufacturer specified absolute maximum limit, as demonstrated previously²⁶.

Comment 2

Lines 127: I assume this should be Fig.2b?

No, “Fig.2d” is correct. This subpanel shows the case in which the DoV and the SNR are uncompromised but the acquisition time is compromised, i.e. increased, in the multi-wavelength TD approach.

Comment 3

Line 259 FCOT can produce depth resolve images (see Suppl FigS2). However, I cannot find these there. Unless this should be FigS3. I propose to use in these images a vertical scale (1 mm?)

Depth-resolve images of human skin are shown in the new Supplementary Figure S2. The whole vertical size is 1 mm, which is also explained in the figure description.

Comment 4

Line 419,420, 468 etc. use μ instead of u

We have replaced u with μ in the text.

Lines 474-475:

Image acquisition occurs in a large Field of View (10x10 mm²) and with a scanning step size of 10 μ m in order to be much lower than the lateral resolution of the system.

Line 490-491:

The spatial resolution is given by the smallest radius at which all the lines can be well resolved (Fig.S3h), which is 38 μ m for all wavelengths.

Line 516-523:

Using the above mentioned values (Table 1) for each laser diode and a scanning speed of 10mm/s for a scanning step size of 10 μ m we can calculate the sample exposure. The total exposure of the sample for simultaneous illumination with all four wavelengths is calculated to be 19.8 μ J/cm² per pulse and 3.96W/cm² mean exposure, well below the MPE limits of 20mJ/cm² and 18W/cm², imposed by the American National Standards Institute³⁷. It has been demonstrated that non-linear effects in OA are present at high energy densities above \sim 7mJ/cm²³⁸. The total sample exposure herein was less than 20 μ J/cm², even in the case of simultaneous illumination with all wavelengths. Therefore, the deposited energy in this study was much lower than the threshold for non-linear effects.

Comment 5

FigS1c, details of the scanning head are difficult to see on paper, I propose to enlarge it.

We have increased the scanning head in Fig.5 to the limits allowed by the paper size.

Reviewer 3

This paper presents two very compelling advancements in optoacoustic tomography (OT). The investigators developed a new (to OT) frequency comb paradigm that mixes both frequency domain and time domain approaches to encoding illumination and decoding the response. The

primary use of this is to encode different optical wavelengths to enable better spectroscopy. The critical performance advantage is an improvement in SNR that scales as the square root of the number of optical wavelengths employed. This encoding scheme is coupled with an advancement in replacing solid-state lasers with cheaper pulsed laser diodes. There is an interaction between two ideas and combined together, the laser diode FCOT paradigm provides a practical system capable of high-performance in-vivo imaging. The concept is particularly appealing, as they are borrowing well-developed frequency comb strategies from other disciplines and introducing them to the field of Optoacoustic Tomography, while the laser diodes provide an instrument development strategy that can scale the wavelengths. However, to establish the improvement of FCOT the SNR analysis should involve a more extensive quantitative evaluation of the measurements.

We thank the reviewer for the encouraging comments and for noting the novelty of our work.

Comment 1

While the improvements over of FCOT over time-domain OT can be evaluated through either comparative frame rates or comparative SNR, since the faster imaging is a chosen setting, the performance goal of SNR is the more important feature to validate. While this nominally done in Figure 2 (e.g. 2h), where it is stated that “the SNR of the optical acoustic signal is reduced (for traditional OT)” the results are not clear. While from an \sqrt{N} analytical perspective a 2x increase in SNR would be expected - it is not clear that this was measured in a quantitative analysis of the data. This should be done, and it is likely fairly straightforward.

We would like to thank the reviewer for bringing this to our attention. We have added the calculations of the standard deviation of the noise and the signal intensity to the existing SNR measurements in the manuscript. We have also updated Fig 2 to include enlarged images of the noise level as a figure inset and the SNR values in each case.

Lines 179-181:

The “Noise Level” inset shows a separate measurement of the noise level that yielded a standard deviation of 90.6 μ V. The signal intensity of the OA from LD1 was measured to be 12 mV, resulting in an SNR of 21.2 dB.

Lines 197-202:

In this case, the noise standard deviation (presented in the Figure insets) was 161.6 μ V, which is higher than the 90.6 μ V measured in the pulse train with 200 pulses to average (Fig.2f), while the signal intensity remained 12 mV resulting in an SNR of 18.8 dB for wavelength 1. Finally, with $f_{rep,j}=50$ kHz, $N_p=200$, and $DoV=7.5$ mm for each laser, the OA signal of all 4 lasers can be recorded without SNR losses ($N=90.6$ μ V and $S=12$ mV with 21.2dB for wavelength 1) but with a longer acquisition time, $t_{acq}=4$ ms that is presented in Fig.2i.

Line 210-211:

... SNR (here again $N=90.6 \mu\text{V}$ and $S=12 \text{ mV}$ resulting in an SNR of 21.2 dB for wavelength 1).

We have also discussed why we detect an SNR increase of 2.3dB instead of the theoretical value of 3dB, which I attributed to the existence of systematic noise in our system. This system noise is however present in normal TD averaging and FM technique.

Lines 213-217:

We would expect an SNR increase of 3 dB upon increasing the numbers of averaged measurements from 50 to 200. However, we measured an SNR increase of 2.3 dB for both TD averaging and FM. This discrepancy can be attributed to the presence of systematic electromagnetic noise in the system, which is also averaged and reduces the SNR by 0.7 dB. However, this effect is a general feature of the OA system and not a disadvantage of the FM algorithm.

We have also added a short explanation in the methods section about the SNR formula we used, in Lines 469-470:

To calculate the Signal-to-Noise ratio (SNR) we used the following formula, $10 \cdot \log_{10}(S/N)$, where S is the signal intensity and N the standard deviation of the noise floor.

Comment 2

This comment is an extension of the first. Which feature in figure 2 is demonstrating the improved SNR? What should one look at? Maybe a plot of the residuals would show it better? Not sure. Possibly the feature is the background variance vs the “signal” structure. To these untrained eyes, plots 2f-I all look very similar, with no discernable difference between the methods. Because SNR is the main point of the paper, the visual intuition of the improvement should be much clearer and cleaner.

We have now addressed this in our corrections and additions described in our answers to the previous comments.

Comment 3

SNR analysis is not presented in either figure 3 or 4. For example, Figure 3 is impressive, with very detailed imaging of the ear. But there is no analysis of SNR. Since the main point of the FCOT is SNR, it would seem logical to have an SNR comparison with the in-vivo data also. That also is likely straightforward to implement, possibly with an alternate analysis of the existing data. This analysis is critical, otherwise Figs. 3 and 4 have limited verification/ validation value.

We understand the concerns of the reviewer and we would like to thank them for the encouraging comments regarding the image quality of the mouse ear. However, we do not believe an SNR comparison at the image level will provide additional information. This is because the signals were collected and averaged using the FM algorithm point-by-point before being fed into the reconstruction algorithm. The point-by-point signal processing has already been

demonstrated in Figures 1 and 2. More specifically, we have already proven that TD averaging and FM averaging provide exactly the same signal (Fig 1) and by using the same signals as an input for the reconstruction algorithm would result in the exact same two images as the output. Additionally, we did prove the SNR advantages of FMOT compared to TD multi-wavelength illumination. Simply higher SNR signals as the reconstruction algorithm input would deliver a higher SNR image.

To clarify it we added in the manuscript Lines 262-263:

The data were collected on a grid and were first averaged point-by-point using the FM algorithm before being fed into the reconstruction algorithm (see methods).

We show in Fig 1 and Fig 2 (with relevant improvements in response to the comments of reviewers 1 and 2) that FM delivers the exact same SNR as TD for a single wavelength, and increased SNR or reduced acquisition time in the case of multiple wavelengths. We also demonstrate in Fig. 3 and Fig. S2 (human skin imaging) that there is no cross-talk between the wavelengths.

Comment 4

The discussion of extending up to 28 wavelengths is intriguing because the SNR improvements become more dramatic with increasing n . Since this topic is central to the paper this analysis and prospect should be further developed. In what scenarios would 28 wavelengths be useful? Are there limits to how many frequencies can be encoded.

Following this comment of the reviewer we have improved the corresponding part of the discussion. Lines 368-375:

There is also a limit to the maximum number of wavelengths used in FMOT that depends on the UST detection bandwidth, the acquisition time, or the number of averages in the pulse trains and the repetition rate of L1 used in each case. When $f_{rep,1}=200$ kHz, $N_p=100$, and the UST bandwidth between 22-78MHz, the maximum number of wavelengths is 28 for FMOT compared to just 5 for TD implementations with the same operating parameters and a DoV of 1.5 mm per wavelength (see Supplementary material). The more wavelengths used in FMOT, the greater the SNR increase per wavelength compared to TD OA. The use of many simultaneous wavelengths would be particularly appealing for improving the molecular detection specificity of OA spectroscopy.

Comment 5

It is easy to accept that all of MRI is in the frequency domain. But is it true that all of OCT is in the frequency domain? Some OCT is swept-source that probably is a bit more like sequential time domain, a few references with clarification would be helpful.

We agree with the reviewer that the statements regarding MRI and OCT needed more nuance. We also think that the comparison between MRI, OCT, and OA was superfluous. Therefore we simplified the arguments while still showing that a switch to FD has benefited both MRI and OCT.

Line 42-44:

Other imaging modalities such as optical coherence tomography (OCT) or magnetic resonance imaging (MRI) were originally demonstrated in the TD, but have benefited, in terms of imaging speed and SNR, from switching operation to the Frequency Domain (FD)^{12,13}.

Comment 6

In Figure 2, the labels of c and d, with the $4 \times N_p/4$ and red circles are difficult to follow. The motivation is understandable. Keeping track of things is tricky because of the interaction between acquisition time, the number of pulses, etc... but if there is a way to improve the clarity that would help many readers.

We would like to thank the reviewer for this comment. The reviewer understands the motivation and the tricky nature of the sacrifices that need to be made in TD multiplexing. We tried to improve the clarity of the figure, removed the red circles that were more of a distraction than help, and overall improve Fig.2.

REVIEWERS' COMMENTS

Reviewer #1 (Remarks to the Author):

The authors properly addressed all of my concerns and I find the current manuscript is very well written. Therefore, it is ready to be published, in my opinion.

Reviewer #3 (Remarks to the Author):

This paper presents a novel new method for frequency Multiplexing Optoacoustic Tomography (FMOT), that solves a problem for concurrently optimizing SNR, Frame rate, and Depth.

The original reviews found the paper noteworthy and significant, with critiques requesting clarification of several concepts and better support of several aspects of the methods and results. The Authors have provided a set of significant and substantial revisions that are detailed below. Overall the paper is now ready for publication.

Specifically,

1. For Reviewer 1: The authors adopted a clearer definition of the technology.
2. Clarified how the optoacoustic signal is generated.
3. Clarified that even with multiple laser diodes the total illumination remains well under the energy required for non-linear effects.
4. There was a comment that actual experimental frequency is not a true delta function but has a linewidth determined at a minimum by the time window. This is clarified in several ways through the main text and supplemental text.
5. In a similar theme, the technique at "heart" relies on the temporal waveforms and Fourier transformed spectra, and these were previously missing from the manuscript. The authors added two supplemental figures that illustrate these concepts and almost a page of supplemental text. This response is sufficient.
6. There was a concern about speckle from the multimode fibers creating un-even illumination. As the reviewers note it is unlikely that the speckles are larger than 38 microns. This seems reasonable. One would need a scenario with a very small effective numerical aperture to generate speckle that big - or would need to relay the speckle pattern with an imaging lens and large magnification. Neither condition is obvious in their setup.

7. They added resolution measures, which is very helpful.
8. A new main text figure better describes how the laser diode illumination works.
9. For reviewer 2 they improved the general audience perspective of the introduction. This revision is extensive and successful.
10. Simple clarification of overdriving lasers by using pulse much much shorter than the thermal time constant of the Laser diodes.
11. A series of copy editing revisions.
12. For Reviewer 3. Extensive revision of the text to clarify SNR which is a main point of the paper.
13. For comments #2————— Reviewer 3 is asking for SNR calculations for the In-vivo data. I understand the challenges of doing this in vivo. However, this seems reasonable... this is the only comment I think that is a little less than completely addressed. They do address this with bench testing.
14. Comment #4 asks about how well the multiple frequency approach scales. This is addressed well, bringing in bandwidth limitations, and is clarified in the text.
15. Comment #5 asks about whether MRI being a k-space approach is a useful reference. This text is improved to provide a more nuanced and accurate statement.
16. Comment #6 asks for better clarity throughout which is provided.

In summary, the authors have made significant and substantial changes to the manuscript including several new figures. There is only one, #13, that could be better addressed. But that is easily greatly overcome by the 12 successful changes.

The paper is ready for publication.

Point-by-Point Responses to Reviewer Comments

We thank the reviewers for thoroughly evaluating our work and providing thoughtful suggestions for improvement. We have endeavored to address as many of the suggestions as possible in our revised manuscript. Below are point-by-point responses to the reviewers' comments. The color code is as follows: reviewer comments are in blue, our responses are in black.

Reviewer 1

The authors properly addressed all of my concerns and I find the current manuscript is very well written. Therefore, it is ready to be published, in my opinion.

We would like to thank reviewer 1 for their helpful comments during last revision and we are happy that we were able to address all of them.

Reviewer 3

This paper presents a novel new method for frequency Multiplexing Optoacoustic Tomography (FMOT), that solves a problem for concurrently optimizing SNR, Frame rate, and Depth. The original reviews found the paper noteworthy and significant, with critiques requesting clarification of several concepts and better support of several aspects of the methods and results. The Authors have provided a set of significant and substantial revisions that are detailed below. Overall the paper is now ready for publication.

Specifically,

1. For Reviewer 1: The authors adopted a clearer definition of the technology.
2. Clarified how the optoacoustic signal is generated.
3. Clarified that even with multiple laser diodes the total illumination remains well under the energy required for non-linear effects.
4. There was a comment that actual experimental frequency is not a true delta function but has a linewidth determined at a minimum by the time window. This is clarified in several ways through the main text and supplemental text.
5. In a similar theme, the technique at "heart" relies on the temporal waveforms and Fourier transformed spectra, and these were previously missing from the manuscript. The authors added two supplemental figures that illustrate these concepts and almost a page of supplemental text. This response is sufficient.
6. There was a concern about speckle from the multimode fibers creating un-even illumination. As the reviewers note it is unlikely that the speckles are larger than 38 microns. This seems reasonable. One would need a scenario with a very small effective numerical aperture to generate speckle that big - or would need to relay the speckle pattern with an imaging lens and

large magnification. Neither condition is obvious in their setup.

7. They added resolution measures, which is very helpful.

8. A new main text figure better describes how the laser diode illumination works.

9. For reviewer 2 they improved the general audience perspective of the introduction. This revision is extensive and successful.

10. Simple clarification of overdriving lasers by using pulse much much shorter than the thermal time constant of the Laser diodes.

11. A series of copy editing revisions.

12. For Reviewer 3. Extensive revision of the text to clarify SNR which is a main point of the paper.

13. For comments #2 Reviewer 3 is asking for SNR calculations for the In-vivo data. I understand the challenges of doing this in vivo. However, this seems reasonable... this is the only comment I think that is a little less than completely addressed. They do address this with bench testing.

14. Comment #4 asks about how well the multiple frequency approach scales. This is addressed well, bringing in bandwidth limitations, and is clarified in the text.

15. Comment #5 asks about whether MRI being a k-space approach is a useful reference. This text is improved to provide a more nuanced and accurate statement.

16. Comment #6 asks for better clarity throughout which is provided.

In summary, the authors have made significant and substantial changes to the manuscript including several new figures. There is only one, #13, that could be better addressed. But that is easily greatly overcome by the 12 successful changes.

The paper is ready for publication.

We would like to thank reviewer 3 for the detailed explanation of our answers to their comments and we are pleased that we have managed to address them.

Comment 13

- For comments #2 Reviewer 3 is asking for SNR calculations for the In-vivo data. I understand the challenges of doing this in vivo. However, this seems reasonable... this is the only comment I think that is a little less than completely addressed. They do address this with bench testing.

We agree with the reviewer that addressing the SNR gains in vivo is reasonable and challenging. However, we already have explained that the FWM algorithm works on the signal processing stage. We have already proven the SNR gains when multiple wavelengths are used with bench testing, as the reviewer acknowledges. We think that it would not significantly enhance the paper to show that feeding the reconstruction algorithm with higher SNR signals will deliver higher SNR images. Therefore, we think that the data presented in the manuscript are sufficient.